# The anti-symmetric and anisotropic symmetric exchange interactions between electric dipoles in hafnia

Longju Yu [1], Hong Jian Zhao [1,2,3,4] ✉, Peng Chen [5], Laurent Bellaiche [5] & Yanming Ma [1,3,4] ✉

The anti-symmetric and anisotropic symmetric exchange interactions between two magnetic dipole moments – responsible for intriguing magnetic textures (e.g., magnetic skyrmions) – have been discovered since last century, while their electric analogues were either hidden for a long time or still not known. It is only recently that the anti-symmetric exchange interactions between electric dipoles was proved to exist (with materials hosting such an interaction being still rare) and the existence of anisotropic symmetric exchange interaction between electric dipoles remains ambiguous. Here, by symmetry analysis and first-principles calculations, we identify hafnia as a candidate material hosting the non-collinear dipole alignments, the analysis of which reveals the anti-symmetric and anisotropic symmetric exchange interactions between electric dipoles in this material. Our findings can hopefully deepen the current knowledge of electromagnetism in condensed matter, and imply the possibility of discovering novel states of matter (e.g., electric skyrmions) in hafnia-related materials.

In the last century, the profound exchange couplings between magnetic dipole moments—namely, the magnetic anti-symmetric exchange interactions (also termed as magnetic Dzyaloshinskii–Moriya interaction, mDMI) and the magnetic anisotropic symmetric exchange interaction (mASEI)—were derived with the origin attributed to spin–orbit interaction (see e.g., refs. [1–3]). The mDMI and mASEI are physical underpinnings for many intriguing non-collinear magnetic textures (e.g., magnetic vortices[4,5], skyrmions[6–13], and merons[12–15]) that are promising for novel device applications in information technology[16–18]. Strikingly, electric vortices, skyrmions and merons have also been observed and/or predicted in ferroelectric nanostructures such as Pb(Zr,Ti)O$_3$ nanodisks, nanorods and thin films[19–21], BaTiO$_3$–SrTiO$_3$ nanocomposites[22], PbTiO$_3$ thin films[23], PbTiO$_3$ nanodomains[24] and SrTiO$_3$/PbTiO$_3$ superlattices[25–29]. Unlike the non-collinear magnetic

textures, the mechanisms for these non-collinear dipolar textures were mostly ascribed to the depolarizing field or the Bloch-like domain walls, instead of the exchange interactions between electric dipoles[19,22–29]. To understand the non-collinear ferroelectricity in bulk materials (i.e., with no depolarizing field or domain wall)[30–36], the exchange interactions between electric dipoles were recently revisited, yielding the disclosure of the electric DMI (eDMI)[37–39]—which may also be responsible for the recently observed and so-called double-$Q$-modulated structure[40]. Even so, materials that are known to host eDMI are rather rare and elusive in nature. Besides, whether there is electric anisotropic symmetric exchange interaction (eASEI) is currently ambiguous.

Here, via symmetry analysis and first-principles calculations, we identify hafnia (HfO$_2$) material as an ideal candidate that accommodates the eDMI and eASEI between electric dipoles. We show that

[1]Key Laboratory of Material Simulation Methods and Software of Ministry of Education, College of Physics, Jilin University, Changchun 130012, China. [2]Key Laboratory of Physics and Technology for Advanced Batteries (Ministry of Education), College of Physics, Jilin University, Changchun 130012, China. [3]State Key Laboratory of Superhard Materials, College of Physics, Jilin University, Changchun 130012, China. [4]International Center of Future Science, Jilin University, Changchun 130012, China. [5]Physics Department and Institute for Nanoscience and Engineering, University of Arkansas, Fayetteville, AR 72701, USA. ✉e-mail: physzhaohj@jlu.edu.cn; mym@jlu.edu.cn

$HfO_2$ has various polymorphisms (i.e., $P2_1/c$, $Pmn2_1$, $Pca2_1$, and $Pbca$ phases) demonstrating non-collinear alignments of electric dipoles. The non-collinear dipole patterns (NCDP) herein are interpreted by our phenomenological theories, revealing the existence of eDMI which stems from the structural distortions associated with the oxygen sublattice. We further identify the eAESI in $HfO_2$ (irrelevant to the NCDP), contributed by the oxygen-sublattice structural distortions as well as the long-range and short-range dipolar interactions.

## Results

### The NCDP in HfO₂'s structural phases

Experimentally, $HfO_2$ was found to be polymorphic, with a variety of structural phases such as $Fm\bar{3}m$[41], $P4_2/nmc$[42], $Pbca$[43], $Pnma$[44], $Pbcm$[45], $Pca2_1$[46], and $P2_1/c$[47]. Recent works by first-principles simulations also highlight the possibility of achieving the polar $Pmn2_1$ phase of $HfO_2$ (see, e.g., refs. 48–50). Of particular interest are the $P2_1/c$, $Pmn2_1$, $Pca2_1$ and $Pbca$ phases. As will be shown below, these phases exhibit NCDP, and analyzing these NCDP enables the disclosure of the eDMI and eASEI in $HfO_2$. In the following, we represent the electric dipoles in structural phases of $HfO_2$ by the displacements of Hf ions, with respect to their positions in the reference structure. We shall show that the non-collinear alignments of dipoles in $P2_1/c$, $Pmn2_1$, $Pca2_1$, and $Pbca$ phases can be well understood by investigating the structural distortions of $HfO_2$. The possible structural distortions in $HfO_2$ are described in Fig. 1 with the conventional cell of $Fm\bar{3}m$ $HfO_2$ being selected as our reference structure. The reasons for such a selection are as follows. First, our symmetry analysis based on the conventional cell of $Fm\bar{3}m$ $HfO_2$ can well describe the NCDP in $P2_1/c$, $Pmn2_1$, $Pca2_1$ and $Pbca$ phases. Second, using a larger cell, although capturing more abundant

structural distortions and NCDP, will significantly increase the difficulties for our symmetry analysis.

Now, we analyze the possible structural distortions accommodated by the high-symmetric $Fm\bar{3}m$ phase of $HfO_2$, prior to extracting the NCDP in $P2_1/c$, $Pmn2_1$, $Pca2_1$ and $Pbca$ phases. As shown in Fig. 1a, the conventional cell of $Fm\bar{3}m$ $HfO_2$ is composed of two sublattices made of Hf ions (see Fig. 1b) and O ions (see Fig. 1c). The Hf sublattice hosts four types of lattice modes sketched in Fig. 1d and labeled by $Hf^U$ ($U = F, X, Y, Z$). By linking the $Hf^U$ mode with $u_\alpha$ [i.e., atomic displacement along $\alpha$ direction ($\alpha = x, y, z$)], we arrive at the structural distortion mode $Hf_\alpha^U$—termed as "order parameter" in the following. For example, the definitions of $Hf_x^X$, $Hf_y^X$ and $Hf_z^X$ order parameters are depicted by the red dash arrow, yellow solid arrow, and purple dot arrow, respectively (Fig. 1e). Similarly, we can define the other order parameters associated with Hf sublattice (Fig. 1d) and those contributed by the O sublattice (Fig. 1f) in a self-explanatory manner. In this regard, the order parameters associated with Hf and O sublattices are symbolized as $Hf_\alpha^U$ and $O_\gamma^W$, respectively. Here, the superscript $U$ or $W$ indicates the lattice mode (Fig. 1d, f) and the subscript $\alpha$ or $\gamma$ marks the direction of the atomic displacements. Following this convention, we have identified thirty-six order parameters for $HfO_2$ [see Supplementary Note 1 in the Supplementary Information (SI) for details].

Starting from these thirty-six order parameters, we construct phenomenological theories that describe the NCDP in $HfO_2$. We notice that the combination of $Hf_\alpha^U$ and $Hf_\beta^V$ order parameters naturally yields NCDP, when $U \neq V$ and $\alpha \neq \beta$ (see Fig. 1g–i). By symmetry arguments, $Hf_\alpha^U$ and $Hf_\beta^V$ are possibly coexisting via the $Hf_\alpha^U Hf_\beta^V O_\gamma^W$ trilinear coupling, that mediated by the $O_\gamma^W$ structural order parameter. As shown in Supplementary Note 1 of the SI, we have derived four effective

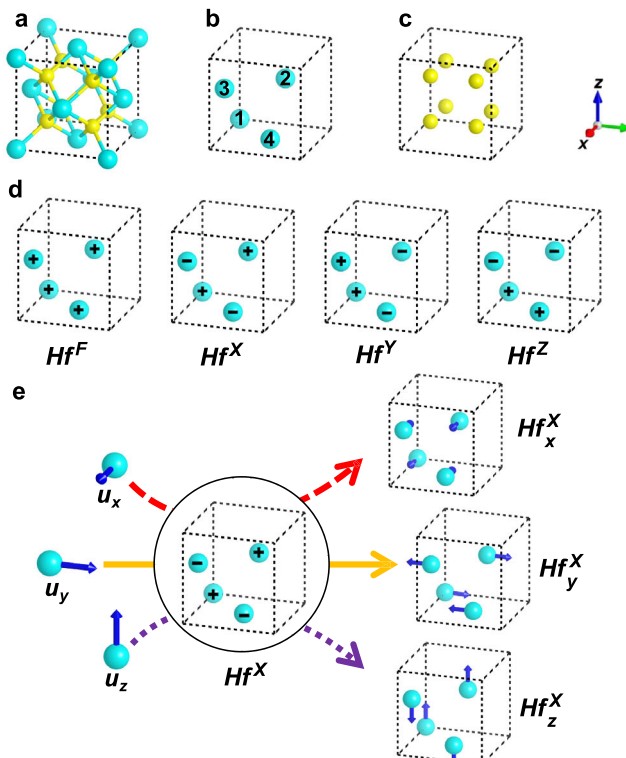

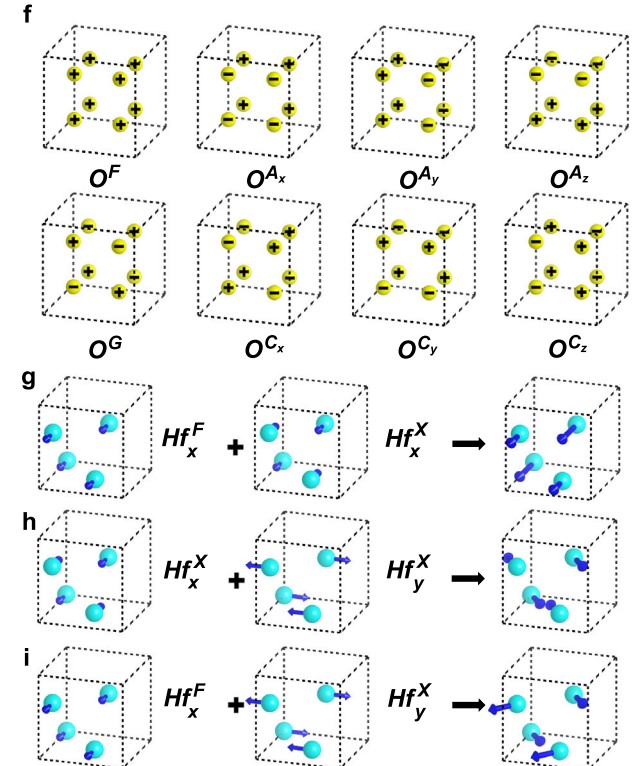

**Fig. 1 | The possible structural distortions in hafnia with respect to its cubic phase. a** The conventional cell of the cubic $Fm\bar{3}m$ phase. **b, c** The sublattices made of Hf and O ions. **d** Four lattice modes associated with the Hf sublattice. **e** Definitions for the $Hf_\alpha^X$ structural distortions. The combination of $Hf^X$-type lattice mode and atomic displacement $u_\alpha$ ($\alpha = x, y, z$) yields the $Hf_\alpha^X$ structural distortions. **f** Eight lattice modes associated with the O sublattice. **g** The collinear dipole patterns resulting from the combination of $Hf_x^F$ and $Hf_x^X$ distortions. **h** The collinear

dipole patterns resulting from the combination of $Hf_x^X$ and $Hf_y^X$ distortions. **i** The non-collinear dipole patterns resulting from the combination of $Hf_x^F$ and $Hf_y^X$ distortions. The Hf and O ions are denoted by cyan and yellow spheres. The atomic displacement $u_\alpha$ is represented by blue arrows. In **b**, each Hf ion is numbered by $\tau$ ($\tau = 1, 2, 3, 4$). In **d** and **f**, the "+" and "−" signs indicate that the atomic displacement $u_\alpha$ centered on that corresponding Hf or O site is along the $+\alpha$ and $-\alpha$ directions, respectively.

**Table 1 | Trilinear couplings resulting in non-collinear dipole patterns for various phases of hafnia**

| Hamiltonian | | Phases |
|---|---|---|
| $H_1$ | $\propto$ $\begin{aligned}&Hf_y^Y Hf_x^Z O_y^{A_x} + Hf_x^Y Hf_z^Z O_z^{A_x}\\&+ Hf_x^X Hf_z^Z O_x^{A_y} + Hf_x^Y Hf_z^Z O_z^{A_y}\\&+ Hf_x^X Hf_y^Y O_x^{A_z} + Hf_z^Z Hf_y^Y O_y^{A_z}\end{aligned}$ | $P2_1/c\ (Hf_x^X Hf_z^Z O_x^{A_y})$<br>$Pca2_1\ (Hf_y^Y Hf_z^Z O_z^{A_x})$<br>$Pbca\ (Hf_z^X Hf_y^Y O_y^{A_z})$ |
| $H_2$ | $\propto$ $\begin{aligned}&Hf_y^F Hf_x^X O_y^{A_x} + Hf_x^F Hf_y^X O_x^{A_x}\\&+ Hf_x^F Hf_y^Y O_x^{A_y} + Hf_z^F Hf_y^Y O_z^{A_y}\\&+ Hf_x^F Hf_z^Z O_x^{A_z} + Hf_y^F Hf_z^Z O_y^{A_z}\end{aligned}$ | $Pmn2_1\ (Hf_x^F Hf_y^Y O_x^{A_y}, Hf_z^F Hf_y^Y O_z^{A_y})$<br>$Pca2_1\ (Hf_z^F Hf_y^Y O_z^{A_y})$ |
| $H_3$ | $\propto$ $\begin{aligned}&Hf_y^F Hf_x^X O_y^{A_x} + Hf_x^F Hf_y^X O_x^{A_x}\\&+ Hf_y^F Hf_x^Y O_y^{A_y} + Hf_x^F Hf_y^Y O_x^{A_y}\\&+ Hf_z^F Hf_x^Z O_z^{A_z} + Hf_x^F Hf_z^Z O_x^{A_z}\end{aligned}$ | $Pca2_1\ (Hf_z^F Hf_x^Z O_x^{A_z})$ |
| $H_4$ | $\propto$ $\begin{aligned}&Hf_x^X Hf_y^Z O_y^{A_x} + Hf_y^X Hf_z^Z O_x^{A_x}\\&+ Hf_x^X Hf_z^Y O_x^{A_y} + Hf_x^X Hf_z^Z O_y^{A_y}\\&+ Hf_z^X Hf_x^Y O_x^{A_z} + Hf_y^F Hf_y^Y O_y^{A_z}\end{aligned}$ | $P2_1/c\ (Hf_x^X Hf_y^Z O_z^{A_y})$ |

Here, the definitions of the notations (e.g., $Hf_y^Y$ and $O_y^{A_x}$) are indicated in Fig. 1. The trilinear coupling associated with a specific phase of $HfO_2$ is shown in the parentheses after the space group of that phase.

Hamiltonians ($H_1$–$H_4$) involving trilinear couplings of our aforementioned kind, summarized in Table 1. We can verify the existence of the couplings in $H_l$ ($l = 1 - 4$) by first-principles numerical calculations, using the following strategy (see Methods for details): to verify the $Hf_\alpha^U Hf_\beta^V O_\gamma^W$ coupling, we (i) start from the $Fm\bar{3}m$ phase and impose a structural distortion according to $O_\gamma^W$ with fixed amplitude, (ii) displace Hf ions following $Hf_\beta^V$ mode with varying magnitude, and (iii) measure the first-principles-calculated forces acting on the Hf sublattice and associated with the $Hf_\alpha^U$ mode. The linear relationship between these forces (related to $Hf_\alpha^U$) and the distortion amplitudes (of $Hf_\beta^V$) will corroborate the existence of the $Hf_\alpha^U Hf_\beta^V O_\gamma^W$ coupling. Figure 2 indeed numerically confirms the existence of several selective trilinear couplings, namely, $Hf_z^X Hf_y^Y O_y^{A_z}$, $Hf_z^X Hf_y^Y O_z^{A_z}$, $Hf_z^F Hf_y^Y O_z^{A_y}$ and $Hf_z^F Hf_x^Z O_x^{A_z}$. For instance, the fittings in Fig. 2b—with the $R^2$ (i.e., coefficient of determination) exceeding 0.999—indicate the linear dependence of $Hf_y^Y$ and $Hf_x^Z$ on $Hf_z^F$. The fitting slopes of 14.58 eV Å$^{-2}$ for $Hf_z^F Hf_y^Y O_z^{A_y}$ and 11.04 eV Å$^{-2}$ for $Hf_z^F Hf_x^Z O_x^{A_z}$ show that the $Hf_z^F Hf_y^Y O_z^{A_y}$ and $Hf_z^F Hf_x^Z O_x^{A_z}$ terms contribute unequally in $HfO_2$. Interestingly, our derived $Hf_z^F Hf_x^Z O_x^{A_z}$ coupling coincides with the trilinear coupling that was claimed to drive the ferroelectricity of $Pca2_1$ $HfO_2$ (see ref. 51).

By analyzing the structural distortions in $P2_1/c$, $Pmn2_1$, $Pca2_1$ and $Pbca$ phases, we are able to extract the NCDP in $HfO_2$ and link the NCDP to our derived trilinear couplings. Such detailed analysis can be found in Supplementary Note 2 of the SI. Here, we provide a graphical approach to "visualize" how our theories interpret the NCDP in $HfO_2$ (see Fig. 3). Sketched in Fig. 3a, the $Hf_\alpha^U$ and $Hf_\beta^V$ couple with each other via the intermediate $O_\gamma^W$ distortion. In such sense, the $Hf_\alpha^U$ distortion will lead to $Hf_\beta^V$ (via $O_\gamma^W$) and vice versa—$Hf_\alpha^U$ and $Hf_\beta^V$ coexisting. In the $P2_1/c$ phase, the $Hf_x^X Hf_z^Z O_x^{A_y}$ and $Hf_x^X Hf_z^Z O_z^{A_y}$ trilinear couplings—shown in Table 1—imply the $(Hf_x^X, Hf_z^Z)$ and $(Hf_x^X, Hf_z^Z)$ combinations, respectively, yielding NCDP (Fig. 3b and h). As in Fig. 3c, f and g, the NCDP in $Pca2_1$ phase come from the $(Hf_y^Y, Hf_x^Z), (Hf_y^Y, Hf_x^F)$ and $(Hf_x^F, Hf_z^Z)$ combinations, rooted in the $Hf_y^Y Hf_x^Z O_x^{A_x}, Hf_x^F Hf_y^Y O_z^{A_y}$ and $Hf_z^F Hf_x^Z O_x^{A_z}$ trilinear couplings (see Table 1 as well). Furthermore, the $Hf_x^F Hf_y^Y O_x^{A_y}$ and $Hf_z^F Hf_y^Y O_z^{A_y}$ couplings lead to the NCDP in $Pmn2_1$ phase [via $(Hf_x^F, Hf_y^Y)$ and $(Hf_z^F, Hf_y^Y)$ combinations, see Fig. 3e, f], while the $Hf_z^X Hf_y^Y O_y^{A_z}$ coupling gives rise to the NCDP in the $Pbca$ phase [via $(Hf_z^X, Hf_y^Y)$ combination, see Fig. 3d]. Our aforementioned analysis thus emphasizes the importance of the $Hf_\alpha^U Hf_\beta^V O_\gamma^W$-type of trilinear couplings ($U \neq V$, $\alpha \neq \beta$) towards the NCDP in $HfO_2$'s structural phases. Here, the central structural distortion is $O_\gamma^W$ contributed by the O sublattice, mediating the interaction between $Hf_\alpha^U$ and $Hf_\beta^V$ distortions. In other words, the $O_\gamma^W$-type distortion is the structural origin of the NCDP in $HfO_2$.

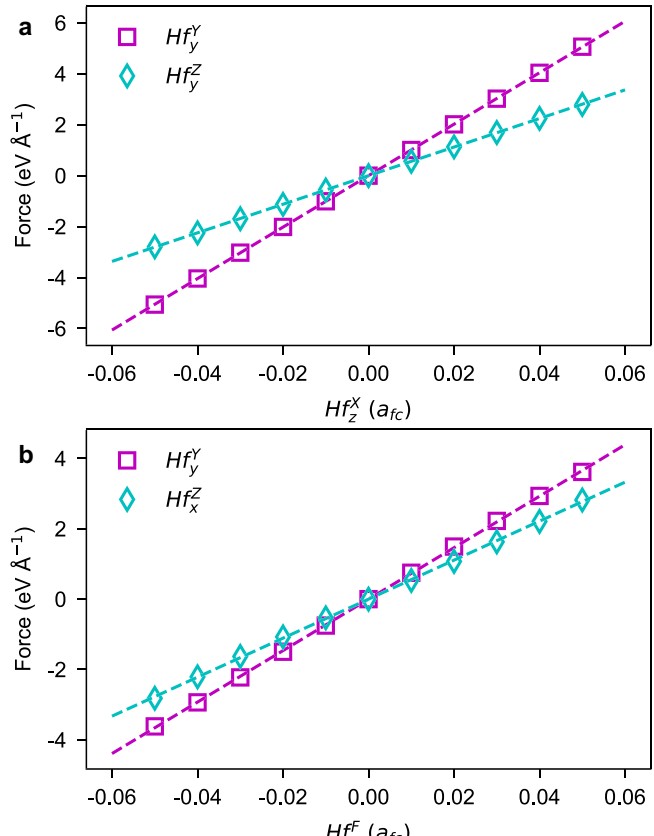

**Fig. 2 | Numerical verification of various trilinear couplings in hafnia. a** Forces on Hf sublattice as a function of $Hf_z^X$ distortion. **b** Forces on Hf sublattice as a function of $Hf_z^F$ distortion. Purple square in (**a**): forces associated with $Hf_y^Y$ mode in $Hf_z^X Hf_y^Y O_y^{A_z}$ ($O_y^{A_z}$ being fixed). Cyan diamond in **a**: forces associated with $Hf_y^Z$ mode in $Hf_z^X Hf_y^Z O_z^{A_y}$ ($O_z^{A_y}$ being fixed). Purple square in **b**: forces associated with $Hf_y^Y$ mode in $Hf_z^F Hf_y^Y O_z^{A_y}$ ($O_z^{A_y}$ being fixed). Cyan diamond in (**b**): forces associated with $Hf_x^Z$ mode in $Hf_z^F Hf_x^Z O_x^{A_z}$ ($O_x^{A_z}$ being fixed). The dash lines in **a** and **b** display the linear fitting results.

## The $O_\gamma^W$-contributed anti-symmetric exchange interactions

The correlation between $Hf_\alpha^U Hf_\beta^V O_\gamma^W$ couplings ($U \neq V$, $\alpha \neq \beta$) and NCDP opens a door to reveal the eDMI and eASEI in $HfO_2$ oxide. In this section, we concentrate on the anti-symmetric eDMI. We recall that the magnetic exchange interaction is given by[3,10]

$$\mathcal{H} = \sum_{i \neq j, \alpha, \beta} J_{ij,\alpha\beta} m_{i,\alpha} m_{j,\beta}, \tag{1}$$

where $m_{i,\alpha}$ and $m_{j,\beta}$ ($\alpha, \beta = x, y, z$) are $\alpha$- and $\beta$-component of magnetic dipole moments centered on the $i$th and $j$th ions, respectively, and $J_{ij,\alpha\beta}$ characterizes the strength of coupling between $m_{i,\alpha}$ and $m_{j,\beta}$. Equation (1) implies that the electric exchange interaction between $u_{i,\alpha}$ and $u_{j,\beta}$ dipoles (if it exists) can be written as

$$H = \sum_{i \neq j, \alpha, \beta} J'_{ij,\alpha\beta} u_{i,\alpha} u_{j,\beta}. \tag{2}$$

Here, $u_{i,\alpha}$ and $u_{j,\beta}$ are atomic displacements, depicting the electric dipoles centered on $i$th and $j$th ions.

We refer interested readers to Supplementary Note 3 of the SI for the detailed evaluation of $J'_{ij,\alpha\beta}$ in $HfO_2$. In the following, we simply outline our derivation of $J'_{ij,\alpha\beta}$ and show the important results. We start from the $Fm\bar{3}m$ phase and work with a big supercell made of

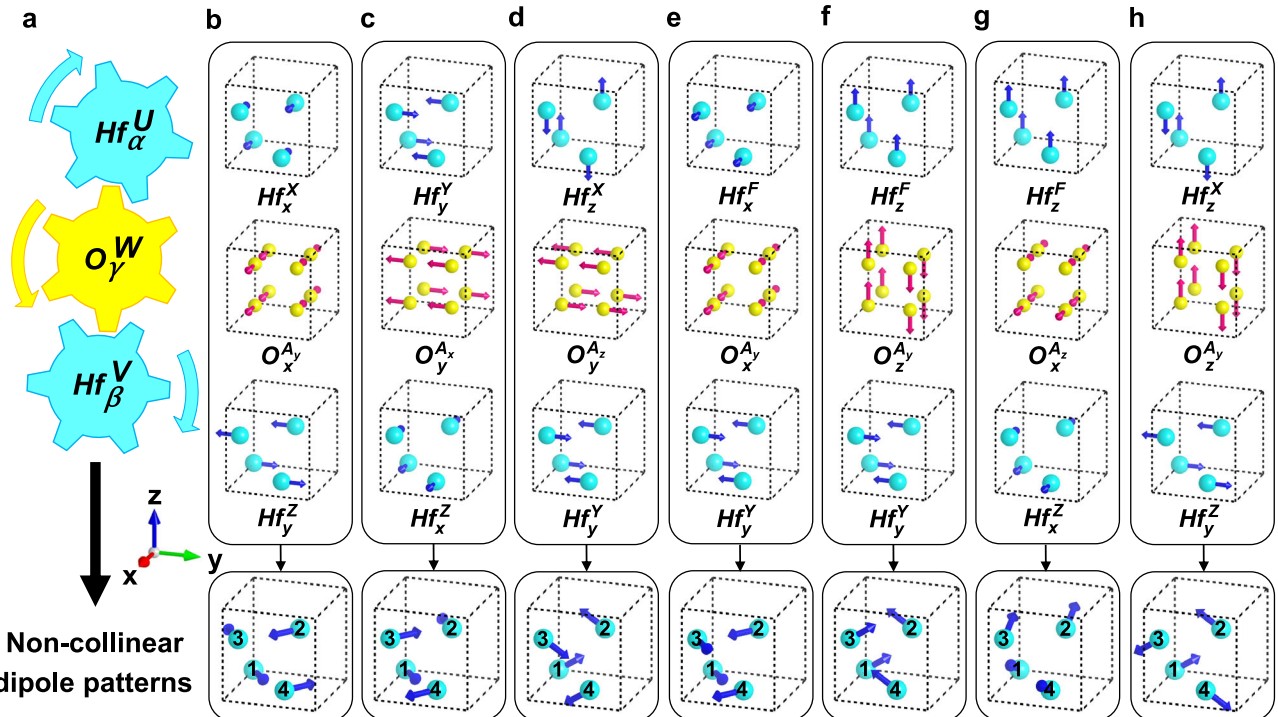

**Fig. 3 | The non-collinear dipole patterns in hafnia resulting from the trilinear couplings. a** Schematic description of the coupling between $Hf_\alpha^U$ and $Hf_\beta^V$ mediated by $O_\gamma^W$. **b-h** The non-collinear dipole patterns associated with $Hf_x^X Hf_y^Z O_x^{A_y}$, $Hf_y^Y Hf_x^Z O_y^{A_x}$, $Hf_z^X Hf_y^Y O_y^{A_z}$, $Hf_x^F Hf_y^Y O_x^{A_y}$, $Hf_z^F Hf_y^Y O_z^{A_y}$, $Hf_z^F Hf_x^Z O_x^{A_z}$ and $Hf_z^X Hf_y^Z O_z^{A_y}$ trilinear couplings. The cyan and yellow spheres denote Hf and O ions, respectively. The atomic displacements are represented by blue or pink arrows.

$N$ conventional cells. Such a supercell contains $4N$ Hf ions with their atomic coordinates given by $\mathbf{R}_m + \mathbf{r}_\tau$, where $\mathbf{R}_m$ locates the $m_{\text{th}}$ conventional cell ($m = 1, 2, \ldots, N$) and $\mathbf{r}_\tau$ is the coordinate of Hf inside the $m_{\text{th}}$ cell ($\tau = 1, 2, 3, 4$, see Fig. 1b). Every Hf ion in the supercell can displace along the $\alpha$ direction ($\alpha = x, y, z$) with respect to $\mathbf{R}_m + \mathbf{r}_\tau$, creating a dipole $u_{m,\tau,\alpha}$ ($m, \tau$ labeling $i$ in $u_{i,\alpha}$). Following Fig. 1b and d, the $Hf_\alpha^F, Hf_\alpha^X, Hf_\alpha^Y$ and $Hf_\alpha^Z$ order parameters can be expanded as

$$Hf_\alpha^F = \tfrac{1}{4N} \sum_m (u_{m,1,\alpha} + u_{m,2,\alpha} + u_{m,3,\alpha} + u_{m,4,\alpha}),$$
$$Hf_\alpha^X = \tfrac{1}{4N} \sum_m (u_{m,1,\alpha} + u_{m,2,\alpha} - u_{m,3,\alpha} - u_{m,4,\alpha}),$$
$$Hf_\alpha^Y = \tfrac{1}{4N} \sum_m (u_{m,1,\alpha} - u_{m,2,\alpha} + u_{m,3,\alpha} - u_{m,4,\alpha}),$$
$$Hf_\alpha^Z = \tfrac{1}{4N} \sum_m (u_{m,1,\alpha} - u_{m,2,\alpha} - u_{m,3,\alpha} + u_{m,4,\alpha}).$$

(3)

Inserting these expansions into $H_l$ ($l = 1 - 4$, see Table 1) yields the effective Hamiltonian as

$$H_l = \sum_{m,m',\tau,\kappa,\alpha,\beta} J'_{m\tau m'\kappa,\alpha\beta} u_{m,\tau,\alpha} u_{m',\kappa,\beta},$$

(4)

where $J'_{m\tau m'\kappa,\alpha\beta}$ – a function of $O_\gamma^W$, $m, m', \kappa, \tau, \alpha$ and $\beta$ – characterizes the coupling between $u_{m,\tau,\alpha}$ and $u_{m',\kappa,\beta}$ dipoles. For instance, the $Hf_x^X Hf_y^Z O_x^{A_y}$ term in $H_1$ implies the coupling between electric dipoles as $\sum_{m,m'} (u_{m,1,x} + u_{m,2,x} - u_{m,3,x} - u_{m,4,x})(u_{m',1,y} - u_{m',2,y} - u_{m',3,y} + u_{m',4,y}) O_x^{A_y}$. By this procedure, we re-formulate each $H_l$ ($l = 1 - 4$) in terms of electric dipole $u_{m,\tau,\alpha}$, and the corresponding $J'_{m\tau m'\kappa,\alpha\beta}$ interaction can be extracted via

$$J'_{m\tau m'\kappa,\alpha\beta} = \frac{\partial^2 H_l}{\partial u_{m,\tau,\alpha} \partial u_{m',\kappa,\beta}}.$$

(5)

By definition, the eDMI between $\mathbf{u}_i \equiv (u_{i,x}, u_{i,y}, u_{i,z})$ and $\mathbf{u}_j \equiv (u_{j,x}, u_{j,y}, u_{j,z})$ dipoles is $\mathbf{D}'_{ij} \cdot (\mathbf{u}_i \times \mathbf{u}_j)$ with $\mathbf{D}'_{ij} = (D'_{ij,x}, D'_{ij,y}, D'_{ij,z})$ being the eDMI vector (see Refs. 37,39). Expanding $\mathbf{D}'_{ij} \cdot (\mathbf{u}_i \times \mathbf{u}_j)$ results in $D'_{ij,x}(u_{i,y} u_{j,z} - u_{i,z} u_{j,y}) + D'_{ij,y}(u_{i,z} u_{j,x} - u_{i,x} u_{j,z}) + D'_{ij,z}(u_{i,x} u_{j,y} - u_{i,y} u_{j,x})$. By $i \to m\tau$ and $j \to m'\kappa$ replacements, such an expansion together with Equations (2)–(5) yield the evaluation of eDMI strength as

$$A'_{m\tau m'\kappa,\alpha\beta} = \frac{1}{2}(J'_{m\tau m'\kappa,\alpha\beta} - J'_{m\tau m'\kappa,\beta\alpha}),$$

(6)

where $D'_{m\tau m'\kappa,x} = A'_{m\tau m'\kappa,yz}$, $D'_{m\tau m'\kappa,y} = A'_{m\tau m'\kappa,zx}$ and $D'_{m\tau m'\kappa,z} = A'_{m\tau m'\kappa,xy}$.

We now explore the eDMI associated with two neighbored Hf ions which belong to the same conventional cell (e.g., $m = m', \tau \neq \kappa$). For the convenience, we omit the cell labels $m$ and $m'$. As for each $H_l$ effective Hamiltonian, the $A'_{\tau\kappa,\alpha\beta}$ components–the eDMI between $Hf_\tau$ and $Hf_\kappa$ pair ($\tau, \kappa = 1, 2, 3, 4$)–form a $3 \times 3$ anti-symmetric matrix with its elements indexed by $\alpha$ and $\beta$. As shown in Supplementary Tables 2, 4, 6 and 8 of the SI, the $A'_{\tau\kappa,\alpha\beta}$ component is determined by the $O_\gamma^W$-type distortion associated with the O sublattice. For example, we examine the interaction involving $Hf_1$ and $Hf_2$ ions, where $\mathbf{r}_\tau \equiv \mathbf{r}_1 = 0$ and $\mathbf{r}_\kappa \equiv \mathbf{r}_2 = 0\mathbf{a} + \frac{1}{2}\mathbf{b} + \frac{1}{2}\mathbf{c}$ ($\mathbf{a}$, $\mathbf{b}$ and $\mathbf{c}$ being the lattice vectors of $Fm\bar{3}m$'s conventional cell). The $H_1$ Hamiltonian suggests that $A'_{12,xy} \propto -O_x^{A_y}$ [see Supplementary Equation (9) and Supplementary Table 2 of the SI]. Similarly, we can extract the eDMI contributed by the $O_x^{A_y}$ distortion, working with a more generalized Hamiltonian $H = \alpha H_1 + \beta H_2 + \gamma H_3 + \delta H_4$. The results are summarized in Table 2. The non-null $A'_{12,xy} = -A'_{12,yx} = \alpha_2 O_x^{A_y}$ (respectively, $A'_{14,xy} = -A'_{14,yx} = \alpha_1 O_x^{A_y}$) components of eDMI imply the non-collinear alignments of electric dipoles–within the $xy$ plane–centered on $Hf_1$ and $Hf_2$ (respectively, $Hf_1$ and $Hf_4$) sites, coinciding with Fig. 3b and e. The detailed analysis confirms that the eDMI drives the NCDP in $P2_1/c$, $Pmn2_1$, $Pca2_1$ and $Pbca$ phases of $HfO_2$ (see Supplementary Note 4 of the SI).

As shown in ref. 39, the $A'_{\tau\kappa,\alpha\beta}$ can be evaluated by $(J'_{\tau\kappa,\alpha\beta} - J'_{\tau\kappa,\beta\alpha})/2$, where $J'_{\tau\kappa,\alpha\beta}$ is the $\alpha\beta$-component of the force

## Table 2 | The $A'_{\tau\kappa}$ and $S'_{\tau\kappa}$ exchange interactions in hafnia

| $(Hf_\tau, Hf_\kappa)$ | $A'_{\tau\kappa,\alpha\beta}$ or $S'_{\tau\kappa,\alpha\beta}$ | $(Hf_\tau, Hf_\kappa)$ | $A'_{\tau\kappa,\alpha\beta}$ or $S'_{\tau\kappa,\alpha\beta}$ | $(Hf_\tau, Hf_\kappa)$ | $A'_{\tau\kappa,\alpha\beta}$ or $S'_{\tau\kappa,\alpha\beta}$ |
|---|---|---|---|---|---|
| $(Hf_1, Hf_2)$ | $A'_{12,xy} = -A'_{12,yx} = \alpha_2 O^{A_y}_x$ | $(Hf_2, Hf_3)$ | $A'_{23,xy} = -A'_{23,yx} = -\alpha_1 O^{A_y}_x$ | $(Hf_3, Hf_4)$ | $A'_{34,xy} = -A'_{34,yx} = \alpha_2 O^{A_y}_x$ |
| $(Hf_1, Hf_3)$ | $S'_{13,xy} = S'_{13,yx} = \alpha_3 O^{A_y}_x$ | $(Hf_2, Hf_4)$ | $S'_{24,xy} = S'_{24,yx} = -\alpha_3 O^{A_y}_x$ | $(Hf_4, Hf_1)$ | $A'_{41,xy} = -A'_{41,yx} = -\alpha_1 O^{A_y}_x$ |
| $(Hf_1, Hf_4)$ | $A'_{14,xy} = -A'_{14,yx} = \alpha_1 O^{A_y}_x$ | $(Hf_3, Hf_1)$ | $S'_{31,xy} = S'_{31,yx} = \alpha_3 O^{A_y}_x$ | $(Hf_4, Hf_2)$ | $S'_{42,xy} = S'_{42,yx} = -\alpha_3 O^{A_y}_x$ |
| $(Hf_2, Hf_1)$ | $A'_{21,xy} = -A'_{21,yx} = -\alpha_2 O^{A_y}_x$ | $(Hf_3, Hf_2)$ | $A'_{32,xy} = -A'_{32,yx} = \alpha_1 O^{A_y}_x$ | $(Hf_4, Hf_3)$ | $A'_{43,xy} = -A'_{43,yx} = -\alpha_2 O^{A_y}_x$ |

The calculations are based on the effective Hamiltonian $H = \alpha H_1 + \beta H_2 + \gamma H_3 + \delta H_4$. In this table, we extract the interactions solely arising from $O^{A_y}_x$ distortion, by setting the other $O^W_\gamma$ as zero. The twelve $Hf_\tau$-$Hf_\kappa$ interactions are marked by $(Hf_\tau, Hf_\kappa)$. Here, we only list the non-null elements of the $A'_{\tau\kappa}$ and $S'_{\tau\kappa}$ tensors (the unlisted elements are zero). The $\alpha_1$, $\alpha_2$ and $\alpha_3$ coefficients are proportional to $\alpha - \beta + \gamma - \delta$, $-\alpha - \beta + \gamma + \delta$ and $-\alpha + \beta + \gamma - \delta$, respectively.

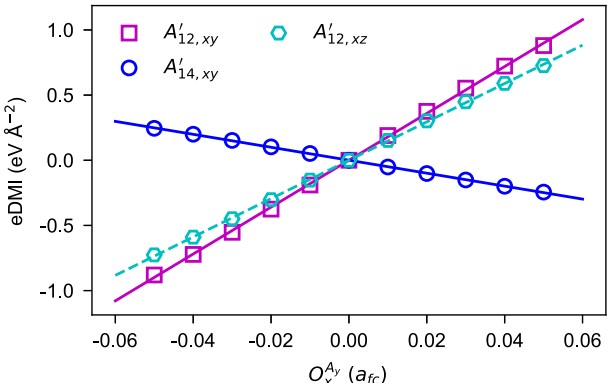

**Fig. 4 | The eDMI in hafnia contributed by the $O^{A_y}_x$ distortion.** The dependence of $A'_{12,xy}$ and $A'_{14,xy}$ components on $O^{A_y}_x$ coincides with $A'_{12,xy} = \alpha_2 O^{A_y}_x$ and $A'_{14,xy} = \alpha_1 O^{A_y}_x$, respectively (see Table 2). The appearance of $A'_{12,xz}$ can be interpreted by a more complicated model (see Supplementary Note 5 of the SI).

constant matrix between $Hf_\tau$ and $Hf_\kappa$ sites [see also Equation (6)]. Now, we quantitatively assess the eDMI in $HfO_2$ using the following strategy. We start from the $Fm\bar{3}m$ $HfO_2$, create the $O^{A_y}_x$ distortion, vary the distortion from $-0.05\,a_{fc}$ to $+0.05\,a_{fc}$, and compute the anti-symmetric exchange interaction $A'_{\tau\kappa,\alpha\beta}$. Figure 4 shows the anti-symmetric exchange interactions $A'_{12,xy}$, $A'_{14,xy}$ and $A'_{12,xz}$ as a function of the $O^{A_y}_x$ distortion. The almost-perfect linear fittings (i.e., $A'_{12,xy}$ versus $O^{A_y}_x$, and $A'_{14,xy}$ versus $O^{A_y}_x$) validate our theories. The fittings also yield that $\alpha_1 = -0.99$ eV Å$^{-3}$ and $\alpha_2 = 3.58$ eV Å$^{-3}$, implying the unequal dependences of $A'_{12,xy}$ and $A'_{14,xy}$ on $O^{A_y}_x$.

Before finishing this section, let us comment on our Table 2 and Fig. 4. Readers may find that Fig. 4 also displays the linear relationship between $A'_{12,xz}$ and $O^{A_y}_x$ distortion, which has not been predicted by our aforementioned theories (see Table 2). This is understandable when realizing that the interactions in Fig. 4 are computed by the finite displacement method, based on the $2 \times 2 \times 2$ supercell (see "Methods"). Enlarging the crystallographic cell will involve more order parameters which implies more abundant couplings. In Supplementary Note 5 of the SI, we show that the anti-symmetric $A'_{12,xz}$ interaction (driven by $O^{A_y}_x$) can come from complicated couplings involving more complex order parameters. Such complex order parameters are redundant to capture our previously discussed NCDP in $P2_1/c$, $Pmn2_1$, $Pca2_1$ and $Pbca$ phases of $HfO_2$. Consequently, we will stop discussing more on these complex order parameters and their couplings.

### The $O^W_\gamma$-contributed anisotropic symmetric exchange interactions

Compared with the mASEI (see e.g., refs. 3,10), the eASEI between $\mathbf{u}_i \equiv (u_{i,x}, u_{i,y}, u_{i,z})$ and $\mathbf{u}_j \equiv (u_{j,x}, u_{j,y}, u_{j,z})$ dipoles (if it exists) can be defined by $\sum_{\alpha\beta} S'_{ij,\alpha\beta} u_{i,\alpha} u_{j,\beta}$, where $\alpha, \beta = x, y, z$, $S'_{ij,\alpha\beta} = S'_{ij,\beta\alpha}$ and

$S'_{ij,xx} + S'_{ij,yy} + S'_{ij,zz} = 0$. Working with Eq. (2) and Eq. (5), the strength of the eASEI between $u_{m,\tau,\alpha}$ and $u_{m',\kappa,\beta}$ is extracted by

$$S'_{m\tau m'\kappa,\alpha\beta} = \frac{1}{2}(J'_{m\tau m'\kappa,\alpha\beta} + J'_{m\tau m'\kappa,\beta\alpha}) \\ - \frac{1}{3}\delta_{\alpha,\beta}(J'_{m\tau m'\kappa,xx} + J'_{m\tau m'\kappa,yy} + J'_{m\tau m'\kappa,zz}), \tag{7}$$

where $\delta_{\alpha,\beta} = 1$ for $\alpha = \beta$ and $\delta_{\alpha,\beta} = 0$ otherwise. The $\alpha\beta$-components of $S'_{m\tau m'\kappa,\alpha\beta}$ form a $3 \times 3$ matrix that is symmetric and traceless. Following $H_1$, $H_2$, $H_3$ and $H_4$, we calculate the eASEI between $\mathbf{u}_{m\tau}$ and $\mathbf{u}_{m\kappa}$ dipoles in $HfO_2$ (see Tables 3, 5, 7, and 9 of the SI). Similar to the eDMI (i.e., $A'_{\tau\kappa,\alpha\beta}$), the eASEI (i.e., $S'_{\tau\kappa,\alpha\beta}$) discussed here are contributed by the $O^W_\gamma$-type distortion as well. However, the dependencies of $A'_{\tau\kappa,\alpha\beta}$ and $S'_{\tau\kappa,\alpha\beta}$ on $O^W_\gamma$ distortion are quite different. To demonstrate this, we consider again the Hamiltonian $H = \alpha H_1 + \beta H_2 + \gamma H_3 + \delta H_4$ and extract the $S'_{\tau\kappa,\alpha\beta}$ associated with $O^{A_y}_x$ (see Table 2). For instance, the $O^{A_y}_x$ distortion results in the non-null eDMI for the $Hf_1$–$Hf_2$, $Hf_1$–$Hf_4$, $Hf_2$–$Hf_3$, and $Hf_3$–$Hf_4$ pairs, while it leads to the non-null eASEI for the $Hf_1$–$Hf_3$ and $Hf_2$–$Hf_4$ pairs. Interestingly, the alignments of the electric dipoles centered on $Hf_1$ and $Hf_3$ sites, linked with the non-null $S'_{13,xy}$, $S'_{13,yx}$, $S'_{31,xy}$ and $S'_{31,yx}$ components, are collinear (see Fig. 3b, e). Our detailed analysis, as shown in Supplementary Note 4 the SI, implies that the eASEI being hosted by $H = \alpha H_1 + \beta H_2 + \gamma H_3 + \delta H_4$ is not relevant to the NCDP in $P2_1/c$, $Pmn2_1$, $Pca2_1$ and $Pbca$ phases of $HfO_2$.

So far, our discussion is based on $Hf^U_\alpha Hf^V_\beta O^W_\gamma$-type couplings ($U \neq V$, $\alpha \neq \beta$) – as indicated in $H_1$, $H_2$, $H_3$, and $H_4$ – that are linked with NCDP. As a by-product, we additionally obtain seven other effective Hamiltonians $H_l$ ($l = 5 - 11$). In contrast to $H_l$ ($l = 1 - 4$), $H_l$ ($l = 5 - 7$) and $H_l$ ($l = 8 - 11$) are effective Hamiltonians with the types of $Hf^U_\alpha Hf^U_\beta O^W_\gamma$ ($\alpha \neq \beta$) and $Hf^U_\alpha Hf^V_\alpha O^W_\gamma$ ($U \neq V$), respectively, being irrelevant to the NCDP in $P2_1/c$, $Pmn2_1$, $Pca2_1$ and $Pbca$ phases of $HfO_2$. As shown in Supplementary Note 3 and Supplementary Tables 10–16 of the SI, these $H_l$ ($l = 5 - 11$) yield the eASEI as well, with the structural origin being the $O^W_\gamma$-type distortion.

### The long-range and short-range interactions

Apart from the exchange interactions mediated by $O^W_\gamma$ distortion, other dipolar interactions can be hosted by $HfO_2$. In ferroelectric theory, the dipolar interaction is written as $\mathcal{H} = \sum_{i \neq j, \alpha\beta} \mathscr{J}_{ij,\alpha\beta} \mu_{i,\alpha} \mu_{j,\beta}$[52], where $\mathscr{J}_{ij,\alpha\beta}$ involves both the long-range and short-range interactions between electric dipoles, and $\mu_{i,\alpha}$ is the amplitude of the local mode centered on the $i_{th}$ cell. Replacing the local mode $\mu_{i,\alpha}$ by our defined $u_{i,\alpha}$, we reach an effective Hamiltonian $\tilde{H} = \sum_{i \neq j, \alpha\beta} \tilde{J}'_{ij,\alpha\beta} u_{i,\alpha} u_{j,\beta}$ and relate the $\tilde{J}'_{ij,\alpha\beta}$ to our aforementioned $J'_{ij,\alpha\beta}$ in Eq. (2). As shown in Table 3, some non-zero components of $\tilde{J}'_{ij,\alpha\beta}$ can appear in high-symmetric $Fm\bar{3}m$ phase of $HfO_2$ (see $\tilde{J}'_{12}$ and $\tilde{J}'_{13}$). Here, the diagonal components

**Table 3 | The dipolar interactions in hafnia**

| $\tilde{J}'_{12}$ (eV Å$^{-2}$) | | | $\tilde{J}'_{13}$ (eV Å$^{-2}$) | | | $\tilde{J}''_{12}$ (eV Å$^{-2}$) | | | $\tilde{J}''_{13}$ (eV Å$^{-2}$) | | |
|---|---|---|---|---|---|---|---|---|---|---|---|
| 3.30 | 0.00 | 0.00 | −2.24 | 0.00 | −2.67 | 3.07 | 0.88 | 0.73 | −2.12 | 0.25 | −2.67 |
| 0.00 | −2.24 | −2.67 | 0.00 | 3.30 | 0.00 | −0.88 | −2.48 | −2.92 | 0.25 | 3.24 | 0.06 |
| 0.00 | −2.67 | −2.24 | −2.67 | 0.00 | −2.24 | −0.73 | −2.92 | −2.45 | −2.67 | 0.06 | −2.26 |

The $\tilde{J}'_{\tau\kappa}$ [$(\tau, \kappa) = (1, 2)$ or $(1, 3)$] matrix characterizes the interactions (between electric dipoles centered on the Hf$_\tau$ and Hf$_\kappa$ sites) for the cubic $Fm\bar{3}m$ phase, while $\tilde{J}''_{\tau\kappa}$ describes those for $Fm\bar{3}m$ phase with a $O_x^{A_y}$ distortion. Here, the magnitude of the $O_x^{A_y}$ distortion is fixed to 0.05 $a_{fc}$, a typical value occurred in the structural phases of HfO$_2$ (see "Methods" for details).

$\tilde{J}'_{12,xx}$ $\tilde{J}'_{12,yy}$ and $\tilde{J}'_{12,zz}$ are 3.30, − 2.24, and − 2.24 eV Å$^{-2}$, respectively. Furthermore, there are also two off-diagonal components, namely, $\tilde{J}'_{12,yz} = \tilde{J}'_{12,zy} = -2.67$ eV Å$^{-2}$. This indicates that the long-range and short-range dipolar interactions can create the eASEI in HfO$_2$, without the participation of $O_\gamma^W$ distortion. In $Fm\bar{3}m$ phase, the $\tilde{J}'_{13}$ is linked with $\tilde{J}'_{12}$ by symmetry (e.g., $\tilde{J}'_{13,yy} = \tilde{J}'_{12,xx}$).

In the presence of $O_x^{A_y}$ structural distortion, the $\tilde{J}'_{12}$ and $\tilde{J}'_{13}$ matrices are modified to $\tilde{J}''_{12}$ and $\tilde{J}''_{13}$, respectively. Now let us make a comparison between $\tilde{J}'_{12}$ and $\tilde{J}''_{12}$. On one hand, the $O_x^{A_y}$ distortion changes the $\tilde{J}'_{12,xx}, \tilde{J}'_{12,yy}, \tilde{J}'_{12,zz}, \tilde{J}'_{12,yz}$ and $\tilde{J}'_{12,zy}$ components. Note that $O_x^{A_y}$ is not the driving force for these components since they originally emerge in the $Fm\bar{3}m$ phase of HfO$_2$. On the other hand, the $O_x^{A_y}$ distortion creates four additional anti-symmetric components (being our aforementioned $O_x^{A_y}$-contributed eDMI). To be specific, $O_x^{A_y}$ distortion with a magnitude of 0.05 $a_{fc}$ induces $\tilde{J}''_{12,xy} = -\tilde{J}''_{12,yx}$ and $\tilde{J}''_{12,xz} = -\tilde{J}''_{12,zx}$ of 0.88 and 0.73 eV Å$^{-2}$, respectively, about 24% of the $\tilde{J}''_{12,xx}$ dipolar interaction. Regarding the $\tilde{J}'_{13}$ and $\tilde{J}''_{13}$ interactions, the $O_x^{A_y}$ distortion induces four additional components (being our aforementioned $O_x^{A_y}$-driven eASEI), namely, $\tilde{J}''_{13,xy} = \tilde{J}''_{13,yx} = 0.25$ eV Å$^{-2}$ and $\tilde{J}''_{13,yz} = \tilde{J}''_{13,zy} = 0.06$ eV Å$^{-2}$. The $\tilde{J}''_{13,xy}$ value is about 8% of the $\tilde{J}''_{13,yy}$ dipolar interaction.

### Exchange interactions: magnetic versus electric

Our previous discussion implies the similarities between the magnetic and electric exchange interactions. This can be further clarified in the following way. In magnetic materials, the magnetic dipole moments are carried by magnetically-active ions (e.g., Fe in LaFeO$_3$). The exchange interaction between magnetic dipole moments $\mathbf{m}_i \equiv (m_{i,x}, m_{i,y}, m_{i,z})$ and $\mathbf{m}_j \equiv (m_{j,x}, m_{j,y}, m_{j,z})$ is given by $\mathcal{H} = \sum_{i\neq j,\alpha,\beta} J_{ij,\alpha\beta} m_{i,\alpha} m_{j,\beta}$ (see e.g., Ref. [10]). Here, $i$ and $j$ characterize the sites of the magnetically-active ions, and $J_{ij,\alpha\beta}$ can be seen as the "force constant" for magnetic dipole moments[39]. The $\mathcal{H}$ interaction is often rewritten as[10]

$$\mathcal{H} = \sum_{i\neq j} J_{ij}^{iso} \mathbf{m}_i \cdot \mathbf{m}_j + \sum_{i\neq j,\alpha\neq\beta} A_{ij,\alpha\beta}(m_{i,\alpha} m_{j,\beta} - m_{i,\beta} m_{j,\alpha}) + \sum_{i\neq j,\alpha,\beta} S_{ij,\alpha\beta} m_{i,\alpha} m_{j,\beta}, \quad (8)$$

where the first term is the Heisenberg exchange interaction, the second term the mDMI, and the third term the mASEI. The Heisenberg exchange parameter $J_{ij}^{iso}$, the mDMI parameter $A_{ij,\alpha\beta}$ and mASEI parameter $S_{ij,\alpha\beta}$ relate to the $J_{ij,\alpha\beta}$ parameter via $J_{ij}^{iso} = (J_{ij,xx} + J_{ij,yy} + J_{ij,zz})/3$, $A_{ij,\alpha\beta} = (J_{ij,\alpha\beta} - J_{ij,\beta\alpha})/2$ and $S_{ij,\alpha\beta} = (J_{ij,\alpha\beta} + J_{ij,\beta\alpha})/2 - \delta_{\alpha,\beta} J_{ij}^{iso}$[10]. In the electric counterpart, the electric dipoles are characterized by the off-center displacements of ferroelectrically-active ions. Such a displacement $\mathbf{u}_i \equiv (u_{i,x}, u_{i,y}, u_{i,z})$ is defined with respect to the equilibrium position of the $i_{th}$ ferroelectrically-active ion in the paraelectric phase. According to lattice dynamics theory, the effective Hamiltonian involving the off-center displacements can be written as $H = \sum_{i\neq j,\alpha,\beta} J'_{ij,\alpha\beta} u_{i,\alpha} u_{j,\beta}$, with $J'_{ij,\alpha\beta}$ being the force constant (of

the paraelectric phase) associated with the $i_{th}$ and $j_{th}$ ferroelectrically-active ions. [This effective Hamiltonian is basically consistent with the ferroelectric theory proposed in ref. [52]. In ref. [52], the electric dipoles are described by local modes (the collective displacements of ions), and the couplings between local modes at different sites include the long-range and short-range interactions]. Reorganizing the $H$ effective Hamiltonian, we arrive at

$$H = \sum_{i\neq j} J_{ij}^{iso} \mathbf{u}_i \cdot \mathbf{u}_j + \sum_{i\neq j,\alpha\neq\beta} A'_{ij,\alpha\beta}(u_{i,\alpha} u_{j,\beta} - u_{i,\beta} u_{j,\alpha}) + \sum_{i\neq j,\alpha,\beta} S'_{ij,\alpha\beta} u_{i,\alpha} u_{j,\beta}, \quad (9)$$

with $J'^{iso}$ being the Heisenberg-like exchange parameter (between $\mathbf{u}_i$ and $\mathbf{u}_j$ dipoles). In this formula, the $A'_{ij,\alpha\beta}$ and $S'_{ij,\alpha\beta}$ are the eDMI and eASEI parameters. This time, the $A'_{ij,\alpha\beta}$ and $S'_{ij,\alpha\beta}$ parameters may be contributed by various factors such as structural distortions but also long-range and short-range dipolar interactions (i.e., not solely by structural distortions). This is readily clarified by comparing $\tilde{J}'_{13}$ with $\tilde{J}''_{13}$ (see Table 3). For example, the $\tilde{J}''_{13,xy}, \tilde{J}''_{13,yx}, \tilde{J}''_{13,yz}$ and $\tilde{J}''_{13,zy}$—for HfO$_2$ with $O_x^{A_y}$ distortion—are driven by the $O_x^{A_y}$ distortion, while $\tilde{J}''_{13,xz}$ and $\tilde{J}''_{13,zx}$ are rooted in the long-range and short-range dipolar interactions. By magnitude, the $\tilde{J}''_{13,xz} = -2.67$ eV Å$^{-2}$ from dipolar interactions is much larger than $\tilde{J}''_{13,xy} = 0.25$ eV Å$^{-2}$ from the $O_x^{A_y}$ distortion. To summarize this paragraph, Eq. (8) and Eq. (9) are quite similar in form, indicating the similarity between magnetic and electric exchange interactions.

Now we discuss the differences between the magnetic and electric exchange interactions. First of all, the magnetic dipole moment $\mathbf{m}_i$ in Eq. (8) is seen as a vector with constant length and varied orientation—a good approximation for treating magnetic insulators, while the electric dipole $\mathbf{u}_i$ in Eq. (9) has both varied length and orientation. Furthermore, the hierarchies of various exchange interactions [e.g., the $(J_{ij}^{iso}, A_{ij,\alpha\beta})$ versus $(J_{ij}'^{iso}, A'_{ij,\alpha\beta})$] are different in the magnetic and electric regimes[53]. For instance, the orders of magnitude for $J_{ij}^{iso}$ ($1 \times 10^{-21}$ J) and $A_{ij,\alpha\beta}$ ($5 \times 10^{-22}$ J) in the magnetic regime are comparable[53]; in sharp contrast, the order of magnitude for electric $J_{ij}'^{iso}$ is $1 \times 10^{-20}$ J, being much larger than $A'_{ij,\alpha\beta}$ ($5 \times 10^{-22}$ J)[53]. In particular, the long-range interaction between magnetic dipoles ($-5 \times 10^{-26}$ J) is much smaller than that ($-1 \times 10^{-20}$ J) between electric dipoles[53]. This seems to interpret the following facts: most of the discovered non-collinear magnetic textures were ascribed to the mDMI, while the mechanisms for the non-collinear dipolar textures were usually ascribed to the depolarizing field rather than eDMI.

### Discussion

Previously, we have demonstrated that electric dipoles, carried by different ferroelectrically-active ions, can couple with each other via electric exchange interactions—being the counterpart of the magnetic exchange interactions. The strength of the coupling between electric dipoles can be evaluated by calculating the interatomic force constants. We are also aware of a recent work focusing on the flexoelectric-like and Dzyaloshinskii−Moriya-like couplings in the continuum Hamiltonian, providing a first-principles approach for determining the various coupling coefficients (including e.g., the eDMI)[54]. Furthermore, we derive the symmetry rules regarding the eDMI and eASEI between electric dipoles (see Supplementary Note 6 of the SI). This allows us to

quickly determine the conditions that prohibit some components of the $A'_{ij}$ and $S'_{ij}$ matrix.

To finish, we show that various structural phases of $HfO_2$ exhibit NCDP. These NCDPs are rooted in the eDMI of electric dipoles. This implies a possible marriage between $HfO_2$-based oxides—high-profile materials in semiconductor technology because of their compatibility with silicon[46,55–64]—and the topological textures of electric dipoles (e.g., electric skyrmions), which are desired states of matter towards the creation of novel information devices[23,25–30,40]. In other words, $HfO_2$ and related materials [e.g., (Hf, Zr)$O_2$ and Y-doped $HfO_2$] may be ideal candidates to explore novel electric topological textures. Besides, we hope that our work can deepen the current knowledge of electromagnetism in condensed matter systems such as ferroelectrics, magnets, and multiferroics.

## Methods
### First-principles simulations
We employ the Vienna Ab-initio Simulation Package (VASP)[65,66] to conduct first-principles simulations. We choose the PBEsol functional[67] based on PAW approach[68] as the exchange-correlation functional. In most of the cases, we work with the conventional cell of $HfO_2$ containing four formula units, using the $12 \times 12 \times 12$ $k$-point mesh for sampling the Brillouin Zone. In other cases, we do computations with respect to the $2 \times 2 \times 2$ supercell of the conventional cell and employ the $6 \times 6 \times 6$ $k$-point mesh. We set the kinetic cutoff energy of 650 eV, solving ($5s$, $5p$, $5d$, $6s$) electrons for Hf and ($2s$, $2p$) electrons for O. For each phase of $HfO_2$, we carry out structural relaxations with the force convergence criterion of 5 meV Å$^{-1}$. In this study, we also use a variety of tools or software—including the Mathematica (https://www.wolfram.com/mathematica), Bilbao Crystallographic Server (https://www.cryst.ehu.es)[69–71] (e.g., AMPLIMODES[72,73] and GENPOS[70]), ISOTROPY Software Suite (https://stokes.byu.edu/iso/isotropy.php) (e.g., FINDSYM[74] and ISODISTORT[75]), VESTA[76], Matplotlib[77]—and the Materials Project database (https://materialsproject.org)[78].

### Numerical verification of trilinear couplings
We use the following strategy to numerically verify our derived $Hf_\alpha^U Hf_\beta^V O_\gamma^W$-type trilinear couplings in $HfO_2$. Starting from the conventional cell of $Fm\bar{3}m$ $HfO_2$ (lattice constant being $a_{fc} = 5.02$ Å), we displace the O ions according to the $O_\gamma^W$ mode by a fixed value of $0.05\,a_{fc}$. Next, we displace the Hf ions following the $Hf_\beta^V$ mode by various values varying from $-0.05\,a_{fc}$ to $0.05\,a_{fc}$ with a step of $0.01\,a_{fc}$. This creates various structures with $O_\gamma^W$ being fixed and $Hf_\beta^V$ being varied. Finally, we do first-principles self-consistent calculations (no structural relaxations) for these distorted structures, measure the resulted forces associated with the $Hf_\alpha^U$ mode, and plot the forces as a function of $Hf_\beta^V$ mode. The numerical results regarding various trilinear couplings are shown in Fig. 2 of the Main Text and Supplementary Fig. 1 of the Supplementary Information.

### The calculation of force constant matrix
We compute the force constant matrix by VASP[65,66] and Phonopy[79,80] using the finite displacement method. During the calculation, the $2 \times 2 \times 2$ supercell (with respect to the conventional cell of $HfO_2$) is used to diminish the interactions between ions and their "images"—arising from the periodic boundary condition. Note that, the $O_\gamma^W$-type distortions in $HfO_2$ are estimated as $(0.06\,a_{fc}, 0.07\,a_{fc})$ for $(O_x^{A_y}, O_y^{A_y})$ in $P2_1/c$ phase, $(0.06\,a_{fc}, 0.06\,a_{fc}, 0.05\,a_{fc})$ for $(O_z^{A_y}, O_y^{A_x}, O_x^{A_z})$ in $Pca2_1$ phase, $0.04\,a_{fc}$ for $O_x^{A_y}$ in $Pmn2_1$ phase, and $0.06\,a_{fc}$ for $O_y^{A_z}$ in $Pbca$ phase. Therefore, we fix the $O_z^{A_y}$ distortion to a typical value (that is, $0.05\,a_{fc}$) for the calculations of $\tilde{J}''_{\tau\kappa}$ interactions in Table 3.

## Data availability
The data that support the findings of this work can be found in the Main Text and the Supplementary Information with the provided source data. Additional information is available by contacting the corresponding authors upon reasonable request. Source data are provided in this paper.

## Code availability
The codes for first-principles-related simulations can be found at https://www.vasp.at/ (VASP) and https://phonopy.github.io/phonopy/ (Phonopy). Other tools or software are available at https://www.cryst.ehu.es/ (AMPLIMODES and GENPOS), https://stokes.byu.edu/iso/isotropy.php/ (ISODISTORT and FINDSYM), https://www.wolfram.com/mathematica/ (Mathematica), https://matplotlib.org/ (Matplotlib), and http://jp-minerals.org/vesta/en/ (VESTA).

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

## Acknowledgements

This work was supported by the National Key Research and Development Program of China (Grant No. 2022YFA1402502), the National Natural Science Foundation of China (Grant No. 12274174, No. 52288102, No. 52090024, and No. 12034009), and the Strategic Priority Research Program of Chinese Academy of Sciences (XDB33000000). P.C. and L.B. thank the Office of Naval Research (ONR) under Grant No. N00014-17-1-2818 and the Vannevar Bush Faculty Fellowship (VBFF) Grant No. N00014-20-1-2834 from the Department of Defense. L.J.Y. acknowledges the support from the high-performance computing center of Jilin University and the support from the International Center of Future Science, Jilin University. The authors thank Prof. M. Alexe, Prof. Y. Nahas, and Prof. S. Prokhorenko for valuable discussion on the eDMI- and eASEI-related phenomena.

## Author contributions

H.J.Z. and Y.M. conceived the project by discussing it with L.B. and P.C. L.J.Y. and H.J.Z. carried out the first-principles calculations and symmetry analysis. All authors contribute to the analysis of the data and the preparation of the paper.

## Competing interests

The authors declare no competing interests.
