## [Peer Review File · Nature Communications]

REVIEWER COMMENTS

Reviewer #1 (Remarks to the Author):

Yu et al propose the existence of an electric Dzyaloshinskii-Moriya Interaction (eDMI) and an electrical Anisotropic Symmetric Exchange Interaction (eASEI) in the various polymorph exhibited by HfO_2 . By analogy with their magnetic counterparts, the eDMI and eASEI can therefore potentially yield several phenomena such as electrical skyrmions or vortices.

The authors use a symmetry analysis to reveal the existence of trilinear terms between several structural distortions, that are further confirmed by first-principles simulations. These types of trilinear couplings were already shown to be coupled to spin canting in oxide perovskite materials, it is really interesting to see that such trilinear couplings also exist in other type of material and can yields novel phenomena based on electric dipoles.

In that sense, I find the paper really interesting with a valuable novelty. Nevertheless, the paper is rather technical and the presentation does not help the understanding with too many calls to supplementary data in the main text. Furthermore, there is no method section in the main text, all informations are "hidden" in the Supplementary. The presentation overall leaves the feeling of a long paper that has been shortened to fit a letter. Although the paper is certainly interesting with no problem with the symmetry analysis and existence of eDMI and eASEI, I am not convinced it fits the broad audience of Nature Communications.

Reviewer #2 (Remarks to the Author):

In this work, the authors focus on the exchange interactions between dipoles in hafnia. By symmetry analysis and DFT calculations, they show that the non-collinear dipole patterns are widely existed in various structural phases of hafnia. The physics of non-collinear dipole patterns is illustrated by deriving the phenomenological energy terms. By analyzing these terms, the authors conclude that the anti-symmetric/symmetric exchange interactions between electric dipoles are existed in hafnia, being the driving force of the non-collinear dipole patterns. This indicates a possibility to realize the electric topological textures (exotic phenonema in the area of ferroelectrics) in hafnia-based materials (star materials in the area of ferroelectrics). Overall, this work is interesting to the ferroelectric and multiferroic communities. I'd like to recommend this work for publication if my following comments are well addressed:

(1) In hafnia, both the eDMI and eASEI are structurally ascribed to O^W_γ -type distortion. Does it mean that the structural origins of eDMI and eASEI are always identical? The authors should clarify this point.

(2) To numerically investigate the effect of eDMI and eASEI on electric dipole patterns, the evaluation of these exchange interaction between dipoles is important. I thus suggest the authors to comment on the approaches for calculating the eDMI and eASEI in various materials.

(3) In magnetism, there are five rules to determine the direction of mDMI by symmetry (see Phys. Rev. 120, 91 (1960) and Natl. Sci. Rev. 6, 629 (2019)). What are the corresponding rules for the eDMI and eASEI?

(4) I notice that the magnetic textures (e.g., magnetic skyrmions) are mostly originated from the mDMI and mASEI, while the electric textures (e.g., electric skyrmions) are basically from another mechanism (i.e., the depolarizing field) rather than the eDMI and eASEI. It will be important to explain why that happens.

Reviewer #3 (Remarks to the Author):

In this paper, the authors try to rationalize theoretically the origin of non-collinear dipolar patterns, addressing the particular case of Hafnia. The topic is relevant and timely and their approach is innovative and interesting. However, I do not find the work suitable for publication in Nature Communications for the following reasons.

Overall, the paper remains technical and sometimes hard to follow. For instance, I appreciate the way the authors introduce the different degrees of freedom in Fig. 1 but they do not properly motivate nor explain what they are doing. The same remains through all along the paper in which many things (like also in the connection between equations) is only said implicitly. All this restricts the interest of the paper to experts of the field who know already what they mean and where they go ... Publishing in a journal like Nature Communications would likely require being more pedagogical.

Then, they claim in their abstract (i) We find that various phases of hafnia showcase non-collinear alignment of electric dipoles, which is interpreted by our phenomenological theories. And (ii) This gives evidence that hafnia simultaneously accommodates anti-symmetric and anisotropic symmetric exchange interactions between electric dipoles. But in fact, it is well known that the different phases of

Hafnia exhibit various non-collinear arrangements of dipoles (as it is true also for other ferroelectrics: i.e. different phases are associated to different arrangements). Then they claim that it gives evidence it accommodates anti-symmetric and anisotropic symmetry exchange: this is also implicitly known since it is the translation of the first evidence in the language of magnetism. What might be new is how to connect the language of interatomic interactions into that of magnetism (anti-symmetric and anisotropic symmetry exchange). Here, their message converges to the key role of trilinear couplings but, which by itself is not new again since it has already been mentioned before (which has likely inspired and motivated their work!). In the present work, it is indeed more interestingly reformulated, in a more practical way, but in itself, this result is rather incremental and this reformulation is finally not fully exploited (i.e. they do not report anything quantitative).

What would be a more significant step forward is to use this reformulation to compute coefficients and exemplify quantitatively the role of the different terms. In fact I am questioning why, in spite of what is reported in Fig. 2, they did not extract and report quantitative parameters !? Are all terms contributing evenly or is there one dominating ? Also how does those trilinear terms compare to other terms in the energy ? It should be noticed that other kinds of terms (than trilinear) could contribute to J and then A and S and there is no guarantee that focusing on the trilinear ones properly capture the dominant part. This has to be done applying a quantitative model up to the end ...

In conclusion, I consider it is a potentially very interesting paper but, reaching its end, I am a bit disappointed. I have the feeling that they stopped half-way. They propose an interesting approach; now they should apply it to quantify things better if they really want to make a step forward. For these reasons, I am not recommending this manuscript at this stage.

Response Letter

First of all, we are grateful to Reviewers #1, #2 and #3 for their very constructive comments and suggestions. We believe that addressing their comments significantly improved the quality of our work. In the following, we provide the point-to-point response to these comments and suggestions.

Reviewer #1 has the following comments:

(0) “Yu et al propose the existence of an electric Dzyaloshinskii-Moriya Interaction (eDMI) and an electrical Anisotropic Symmetric Exchange Interaction (eASEI) in the various polymorph exhibited by HfO_2 . By analogy with their magnetic counterparts, the eDMI and eASEI can therefore potentially yield several phenomena such as electrical skyrmions or vortices. The authors use a symmetry analysis to reveal the existence of trilinear terms between several structural distortions, that are further confirmed by first-principles simulations. These types of trilinear couplings were already shown to be coupled to spin canting in oxide perovskite materials, it is really interesting to see that such trilinear couplings also exist in other type of material and can yields novel phenomena based on electric dipoles. In that sense, I find the paper really interesting with a valuable novelty.”

Authors' response:

We appreciate very much that Reviewer #1 explicitly stated that our work is “really interesting with a valuable novelty”.

(1) “Nevertheless, the paper is rather technical and the presentation does not help the understanding with too many calls to supplementary data in the main text. Furthermore, there is no method section in the main text, all informations are "hidden" in the Supplementary.

The presentation overall leaves the feeling of a long paper that has been shortened to fit a letter. Although the paper is certainly interesting with no problem with the symmetry analysis and existence of eDMI and eASEI, I am not convinced it fits the broad audience of Nature Communications.”

Authors’ response:

We totally agree with Reviewer #1 that efforts should be made to improve the readability of our work, in order to be more beneficial to the broader audience. [Please note that this comment coincides with the point (2) of Reviewer #3.] Following this comment, we have revised our manuscript substantially, with the revisions summarized as follows.

(i) We decided to demonstrate the definition of various order parameters pedagogically. To this end, we reorganized our Fig. 1 and illustrate step-by-step how to arrive at the thirty-six order parameters. For example, we show that the high-symmetric phase of hafnia has two sublattices made of Hf and O ions (Figs. 1a, 1b, 1c), where the Hf sublattice hosts four lattice modes (Fig. 1d) and O sublattice accommodates eight modes (Fig. 1f). Then, we use Fig. 1e to demonstrate that the combination of a specific lattice mode with atomic displacement – along **x**, **y** or **z** direction – further yields three structural distortions (termed as order parameters in this work). We also show that the order parameters associated with the O sublattice can be defined in a similar way. This will help readers to understand the definition of the thirty-six order parameters (for hafnia) without accessing to the Supplementary Information.

(ii) In Figs. 1g-i, we use three sketches to show why the $Hf_{\alpha}^U Hf_{\beta}^V O_{\gamma}^W$ ($U \neq V$, $\alpha \neq \beta$) type couplings lead to non-collinear dipole patterns while the other types [*i.e.*, $Hf_{\alpha}^U Hf_{\beta}^V O_{\gamma}^W$ ($U = V$ or $\alpha = \beta$)] cannot yield them. This explains why we are particularly interested in the $Hf_{\alpha}^U Hf_{\beta}^V O_{\gamma}^W$ ($U \neq V$, $\alpha \neq \beta$) type couplings. Focusing on these couplings, we discover four Hamiltonians that are shown in Table 1 of the Main Text.

(iii) In Supplementary Note 1 and Supplementary Note 2, we provide the detailed analysis regarding Table 1 of the Main Text. To be specific, we show how the four Hamiltonians (in Table 1)

are related to the non-collinear dipole patterns in the $P2_1/c$, $Pca2_1$, $Pmn2_1$ and $Pbca$ phases of hafnia. Since we understand that such analysis may be tedious for some readers, we also provide a graphical approach to “visualize” the connection between these Hamiltonians and the non-collinear dipole patterns in hafnia (see Fig. 3 in the Main Text). In Fig. 3a, we use a sketch to depict that the Hf_α^U and Hf_β^V modes interact with each other via the O_γ^W distortion, where the combination of Hf_α^U and Hf_β^V modes yield the non-collinear dipole patterns. In Figs. 3b-h, we demonstrate how each O_γ^W distortion creates the non-collinear dipole patterns (via trilinear couplings in Table 1) in $P2_1/c$, $Pca2_1$, $Pmn2_1$ and $Pbca$ phases of hafnia.

(iv) In the “The O_γ^W -contributed anti-symmetric exchange interactions” section, we outline our derivation of the anti-symmetric exchange interaction. In particular, we move some formula from the Supplementary Information to the Main Text, to avoid unnecessary calls to the Supplementary Information during the reading of our manuscript.

(v) Furthermore, we have made significant efforts to smoothly connect various contents in our manuscript. Please see our “Authors’ response” to the point (2) of the Reviewer #3 for details.

(vi) We move the Methods section from the Supplementary Information to our Main Text.

In response to this comment, our revisions are indicated below:

-- We have redrawn our Fig. 1 and Fig. 3.

-- In page 2 of the Main Text, we wrote “As shown in Fig. 1a, the conventional cell of $Fm\bar{3}m$ HfO_2 is composed of two sublattices made of Hf ions (see Fig. 1b) and O ions (see Fig. 1c). The Hf sublattice hosts four types of lattice modes sketched in Fig. 1d and labelled by Hf^U ($U = F, X, Y, Z$). By linking the Hf^U mode with u_α [*i.e.*, atomic displacement along α direction ($\alpha = x, y, z$)], we arrive at the structural distortion mode Hf_α^U – termed as “order parameter” in the following. For example, the definitions of Hf_x^X , Hf_y^X and Hf_z^X order parameters are depicted by the red dash arrow, yellow solid arrow, and purple dot arrow, respectively (Fig. 1e). Similarly, we can define the other order parameters associated with Hf sublattice (Fig. 1d) and those contributed by the O sublattice (Fig. 1f) in a self-explanatory manner. In this regard, the order parameters associated with

Hf and O sublattices are symbolized as Hf_{α}^U and O_{γ}^W , respectively. Here, the superscript U or W indicates the lattice mode (Figs. 1d and 1e) and the subscript α or γ marks the direction of the atomic displacements.”

-- In page 3 of the Main Text, we wrote “By analyzing the structural distortions in $P2_1/c$, $Pmn2_1$, $Pca2_1$ and $Pbca$ phases, we are able to extract the NCDP in HfO_2 and link the NCDP to our derived trilinear couplings. Such detailed analysis can be found in Supplementary Note 2 of the SI. Here, we provide a graphical approach to “visualize” how our theories interpret the NCDP in HfO_2 (see Fig. 3). Sketched in Fig. 3a, the Hf_{α}^U and Hf_{β}^V couple with each other via the intermediate O_{γ}^W distortion. In such sense, the Hf_{α}^U distortion will lead to Hf_{β}^V (via O_{γ}^W) and *vice versa* – Hf_{α}^U and Hf_{β}^V coexisting. In the $P2_1/c$ phase, the $Hf_x^X Hf_y^Z O_x^{Ay}$ and $Hf_z^X Hf_y^Z O_z^{Ay}$ trilinear couplings – shown in Table 1 – imply the (Hf_x^X, Hf_y^Z) and (Hf_z^X, Hf_y^Z) combinations, respectively, yielding NCDP (Figs. 3b and 3h).”

-- In page 4 of the Main Text, we wrote “We refer interested readers to Supplementary Note 3 of the SI for the detailed evaluation of $J'_{ij,\alpha\beta}$ in HfO_2 . In the following, we simply outline our derivation of $J'_{ij,\alpha\beta}$ and show the important results.” and “Following Figs. 1b and 1d, the Hf_{α}^F , Hf_{α}^X , Hf_{α}^Y , and Hf_{α}^Z order parameters can be expanded as” followed by an Equation showing the expansion of these order parameters by atomic basis.

We believe that these revisions will be helpful for readers to better understand our work, and are also beneficial to the broad audience. We therefore hope that our work can be recommended for publication in *Nature Communications*.

Reviewer #2 has the following comments:

(0) “In this work, the authors focus on the exchange interactions between dipoles in hafnia. By symmetry analysis and DFT calculations, they show that the non-collinear dipole patterns are widely existed in various structural phases of hafnia. The physics of non-collinear dipole patterns is illustrated by deriving the phenomenological energy terms. By analyzing these terms, the authors conclude that the anti-symmetric/symmetric exchange interactions between electric dipoles are existed in hafnia, being the driving force of the non-collinear dipole patterns. This indicates a possibility to realize the electric topological textures (exotic phenonema in the area of ferroelectrics) in hafnia-based materials (star materials in the area of ferroelectrics). Overall, this work is interesting to the ferroelectric and multiferroic communities. I'd like to recommend this work for publication if my following comments are well addressed:”

Authors' response:

We thank Reviewer #2 for the positive evaluation of our work. In the following, we have addressed all the comments raised by Reviewer #2. We thus hope that our work will be recommended for publication in *Nature Communications*.

(1) “In hafnia, both the eDMI and eASEI are structurally ascribed to O^W_γ -type distortion. Does it mean that the structural origins of eDMI and eASEI are always identical? The authors should clarify this point.”

Authors' response:

We thank Reviewer #2 for this interesting comment. This motivates us to explore the dependence of eDMI and eASEI on the O^W_γ distortion in details. To this end, we consider a more generalized Hamiltonian $H = \alpha H_1 + \beta H_2 + \gamma H_3 + \delta H_4$, relevant to the non-collinear dipole

alignments. Starting from H , we extract the eDMI and eASEI contributed by various O_Y^W distortions. The results are summarized in Table 2 of the Main Text and Tables 17, 18, 19, 20 of the Supplementary Information. The conclusion is that the dependencies of the eDMI $A'_{\tau\kappa,\alpha\beta}$ and eASEI $S'_{\tau\kappa,\alpha\beta}$ on O_Y^W are not identical, although both eDMI and eASEI can be ascribed to O_Y^W .

In response to this comment, we wrote “Similarly, we can extract the eDMI contributed by the O_x^{Ay} distortion, working with a more generalized Hamiltonian $H = \alpha H_1 + \beta H_2 + \gamma H_3 + \delta H_4$. The results are summarized in Table 2. The non-null $A'_{12,xy} = -A'_{12,yx} = \alpha_2 O_x^{Ay}$ (respectively, $A'_{14,xy} = -A'_{14,yx} = \alpha_1 O_x^{Ay}$) components of eDMI imply the non-collinear alignments of electric dipoles – within the xy plane – centered on Hf₁ and Hf₂ (respectively, Hf₁ and Hf₄) sites, coinciding with Figs. 3b and 3e.” in page 5 of the Main Text, and “Similar to the eDMI (*i.e.*, $A'_{\tau\kappa,\alpha\beta}$), the eASEI (*i.e.*, $S'_{\tau\kappa,\alpha\beta}$) discussed here are contributed by the O_Y^W -type distortion as well. However, the dependencies of $A'_{\tau\kappa,\alpha\beta}$ and $S'_{\tau\kappa,\alpha\beta}$ on O_Y^W distortion are quite different. To demonstrate this, we consider again the Hamiltonian $H = \alpha H_1 + \beta H_2 + \gamma H_3 + \delta H_4$ and extract the $S'_{\tau\kappa,\alpha\beta}$ associated with O_x^{Ay} (see Table 2). For instance, the O_x^{Ay} distortion results in the non-null eDMI for the Hf₁-Hf₂, Hf₁-Hf₄, Hf₂-Hf₃ and Hf₃-Hf₄ pairs, while it leads to the non-null eASEI for the Hf₁-Hf₃ and Hf₂-Hf₄ pairs.” in page 6 of the Main Text.

(2) “To numerically investigate the effect of eDMI and eASEI on electric dipole patterns, the evaluation of these exchange interaction between dipoles is important. I thus suggest the authors to comment on the approaches for calculating the eDMI and eASEI in various materials.”

Authors' response:

This is an excellent point as well. In our work, we use the displacements of the ferroelectrically-active ions to represent the electric dipoles. In such sense, the exchange interactions between electric dipoles are related to the interatomic force constants, as suggested by Equation (2) and Ref. [39] of

our Main Text. Besides, a very recent work (Ref. [54] in our Main Text) discussed the flexoelectric-like and Dzyaloshinskii-Moriya-like couplings in the continuum Hamiltonian. In this work, the author provides a first-principles approach for the evaluation of the corresponding coupling coefficients (including *e.g.*, the eDMI).

In response to this comment, we wrote “As shown in Ref. ³⁹, the $A'_{\tau\kappa,\alpha\beta}$ can be evaluated by $(J'_{\tau\kappa,\alpha\beta} - J'_{\tau\kappa,\beta\alpha})/2$, where $J'_{\tau\kappa,\alpha\beta}$ is the $\alpha\beta$ -component of the force constant matrix between Hf_τ and Hf_κ sites [see also Equation (6)]” in page 5 of the Main Text. We also wrote “The strength of the coupling between electric dipoles can be evaluated by calculating the interatomic force constants. We are also aware of a recent work focusing on the flexoelectric-like and Dzyaloshinskii-Moriya-like couplings in the continuum Hamiltonian, providing a first-principles approach for determining the various coupling coefficients (including *e.g.*, the eDMI)⁵⁴.” in the “Discussion” section (page 8) of the Main Text. Furthermore, we demonstrate how to extract the eDMI from the interatomic force constants (see Fig. 4 in the Main Text and the related discussion).

(3) “In magnetism, there are five rules to determine the direction of mDMI by symmetry (see Phys. Rev. 120, 91 (1960) and Natl. Sci. Rev. 6, 629 (2019)). What are the corresponding rules for the eDMI and eASEI?”

Authors’ response:

This is an interesting point too. Definitely, providing such symmetry rules will be helpful for determining the existence or not of some components of eDMI and eASEI (by symmetry) in specific materials without doing numerical simulations. In Supplementary Note 6, we derive the symmetry rules regarding the eDMI and eASEI.

In response to this comment, we wrote “Furthermore, we derive the symmetry rules regarding the eDMI and eASEI between electric dipoles (see Supplementary Note 6 of the SI). This allows us

to quickly determine the conditions that prohibit some components of the A'_{ij} and S'_{ij} matrix.” in page 8 of the Main Text (the “Discussion” section). We also added a “Supplementary Note 6” in our Supplementary Information.

(4) “I notice that the magnetic textures (e.g., magnetic skyrmions) are mostly originated from the mDMI and mASEI, while the electric textures (e.g., electric skyrmions) are basically from another mechanism (i.e., the depolarizing field) rather than the eDMI and eASEI. It will be important to explain why that happens.”

Authors’ response:

We highly appreciate this point. The fact that the magnetic and electric textures are usually of different origin is quite interesting. To address this point, we decided to analyze the similarities and differences between the electric and magnetic exchange interactions. In the “Exchange interactions: magnetic versus electric” section, we show the Hamiltonians describing the magnetic and electric exchange interactions [see Equation (8) and Equation (9) in the Main Text]. Definitely, the electric and magnetic exchange interactions (Heisenberg, anti-symmetric and anisotropic symmetric) have similar mathematical forms. This seems to imply that the eDMI can naturally drive the electric non-collinear textures, in the same way than the mDMI usually induces the magnetic non-collinear textures. However, the magnetic and electric exchange interactions are essentially different in the following aspect. On one hand, the order of magnitude for mDMI ($\sim 5 \times 10^{-22}$ J) is **comparable to** the Heisenberg exchange interaction ($\sim 1 \times 10^{-21}$ J) between magnetic dipole moments. Hence, the mDMI (favoring the non-collinear alignment of magnetic dipole moments) can play an important role in stabilizing the magnetic textures such as magnetic skyrmions. On the other hand, the order of magnitude for eDMI ($\sim 5 \times 10^{-22}$ J) is **much smaller** than the Heisenberg-like exchange interaction ($\sim 1 \times 10^{-20}$ J) between electric dipoles, in sharp contrast to the magnetic case. As a consequence, the Heisenberg-like exchange interactions (favoring the collinear alignment of dipoles) between electric dipoles are predominant, as compared with the eDMI; the non-collinear dipole textures are not easy

to form in bulk materials. Besides, the long-range interaction between magnetic dipoles ($\sim 5 \times 10^{-26}$ J) is much smaller than that ($\sim 1 \times 10^{-20}$ J) between electric dipoles. This seems to interpret why most of the discovered non-collinear magnetic textures are rooted in mDMI, while the mechanisms for the non-collinear dipolar textures are usually ascribed to the depolarizing field. [Note that the orders of magnitude for our aforementioned exchange interactions were picked up from J. Junquera *et al.*, *Rev. Mod. Phys.* **95**, 025001 (2023), Ref. [53] in our Main Text.]

In response to this comment, we have added a new section in our Main Text entitled “**Exchange interactions: magnetic versus electric**”.

Reviewer #3 has the following comments:

(0) “In this paper, the authors try to rationalize theoretically the origin of non-collinear dipolar patterns, addressing the particular case of Hafnia. The topic is relevant and timely and their approach is innovative and interesting.”

Authors’ response:

We are glad to see that Reviewer #3 finds that “the topic is relevant and timely and their approach is innovative and interesting.”

(1) “However, I do not find the work suitable for publication in Nature Communications for the following reasons.”

Authors’ response:

We appreciate that Reviewer #3 raises a sequence of constructive comments. We believe that addressing these comments are rather helpful to improve the quality of our work. We have made enormous efforts to address these comments, with a point-to-point reply indicated below. We hope that our work will now be recommended for publication in *Nature Communications*.

(2) “Overall, the paper remains technical and sometimes hard to follow. For instance, I appreciate the way the authors introduce the different degrees of freedom in Fig. 1 but they do not properly motivate nor explain what they are doing. The same remains through all along the paper in which many things (like also in the connection between equations) is only said implicitly. All this restricts the interest of the paper to experts of the field who know already what they mean and where they go ... Publishing in a journal like Nature Communications would likely require being more pedagogical.”

Authors' response:

This is an excellent point. We agree that efforts should be made to improve the readability of our work. In this regard, we did the following revisions.

(i) We provide our motivations for introducing the various degrees of freedoms (termed as order parameters in our work). In the section “The NCDP in HfO₂'s structural phases”, we first show that HfO₂ has a variety of polymorphisms, with the $P2_1/c$, $Pca2_1$, $Pmn2_1$ and $Pbca$ phases being of particular interest. Then, we indicate why these four phases are particularly interesting by writing “As will be shown below, these phases exhibit NCDP, and analyzing these NCDP enables the disclosure of the eDMI and eASEI in HfO₂.” and “We shall show that the non-collinear alignments of dipoles in $P2_1/c$, $Pmn2_1$, $Pca2_1$ and $Pbca$ phases can be well understood by investigating the structural distortions of HfO₂.” [NCDP means “non-collinear dipole patterns”]. This naturally allows for the introduction of various order parameters (defined in Fig. 1) to describe the possible structural distortions in HfO₂. In Figs. 1g-i, we use three sketches to show why the $Hf_\alpha^U Hf_\beta^V O_\gamma^W$ ($U \neq V$, $\alpha \neq \beta$) type couplings lead to NCDP while the other types [*i.e.*, $Hf_\alpha^U Hf_\beta^V O_\gamma^W$ ($U = V$ or $\alpha = \beta$)] cannot yield NCDP. This implies that $Hf_\alpha^U Hf_\beta^V O_\gamma^W$ ($U \neq V$, $\alpha \neq \beta$) type couplings may interpret the NCDP in hafnia. Focusing on these couplings, we discover four Hamiltonians that are shown in Table 1 of the Main Text. After that, we provide a graphical approach to illustrate how O_γ^W drives the NCDP in hafnia (see Fig. 3). This emphasizes the central role of O_γ^W for the creation of the NCDP. Following this, we demonstrate how O_γ^W is contributed to the anti-symmetric and anisotropic symmetric exchange interactions in hafnia.

(ii) We also realize that being more pedagogical will benefit a broader audience [this point of Reviewer #3 coincides with point (1) of Reviewer #1]. To improve the readability of our work, we redrew our Fig. 1 regarding the definition of order parameters. Here, we show the detailed procedures for defining the order parameters. Also, we redrew our Fig. 3 to clearly show how the trilinear couplings can yield various NCDP. The details can be found in “Authors' response” to the point (1) of the Reviewer #1.

(iii) Furthermore, we made significant efforts to smoothly connect various contents in our

manuscript. A specific example regarding this point is indicated below. In the original version of our manuscript, we wrote “We expand the Hf_α^F , Hf_α^X , Hf_α^Y , and Hf_α^Z , order parameters by the $u_{m,\tau,\alpha}$ basis, and insert these expansions into our derived $H_l (l=1-4)$, as demonstrated in Eqs. (S1)-(S6) of the SM. This yields the effective Hamiltonian as

$$H_l = \sum_{m,m',\tau,\kappa,\alpha,\beta} J'_{m\tau m'\kappa,\alpha\beta} u_{m,\tau,\alpha} u_{m',\kappa,\beta} \quad (3)$$

where $J'_{m\tau m'\kappa,\alpha\beta}$ – a function of O_γ^W , m , m' , κ , τ , α , and β – characterizes the coupling between $u_{m,\tau,\alpha}$ and $u_{m',\kappa,\beta}$ dipoles. As a result, the $J'_{m\tau m'\kappa,\alpha\beta}$ interaction associated with the H_l Hamiltonian can be extracted via

$$J'_{m\tau m'\kappa,\alpha\beta} = \frac{\partial^2 H_l}{\partial u_{m,\tau,\alpha} \partial u_{m',\kappa,\beta}} \quad (4)$$

and the strength of the eDMI is evaluated by [54]

$$A'_{m\tau m'\kappa,\alpha\beta} = \frac{1}{2} (J'_{m\tau m'\kappa,\alpha\beta} - J'_{m\tau m'\kappa,\beta\alpha}) \quad (5)$$

.”

We realize that readers might have the following questions when reading these sentences.

- (A) What does Equation (3) look like after inserting the Eqs. (S2) into $H_l (l=1-4)$?
- (B) How to arrive at Equation (5)?

In the revised manuscript, we wrote “Following Figs. 1b and 1d, the Hf_α^F , Hf_α^X , Hf_α^Y , and Hf_α^Z order parameters can be expanded as

$$\begin{aligned} Hf_\alpha^F &= \frac{1}{4N} \sum_m (u_{m,1,\alpha} + u_{m,2,\alpha} + u_{m,3,\alpha} + u_{m,4,\alpha}), \\ Hf_\alpha^X &= \frac{1}{4N} \sum_m (u_{m,1,\alpha} + u_{m,2,\alpha} - u_{m,3,\alpha} - u_{m,4,\alpha}), \\ Hf_\alpha^Y &= \frac{1}{4N} \sum_m (u_{m,1,\alpha} - u_{m,2,\alpha} + u_{m,3,\alpha} - u_{m,4,\alpha}), \\ Hf_\alpha^Z &= \frac{1}{4N} \sum_m (u_{m,1,\alpha} - u_{m,2,\alpha} - u_{m,3,\alpha} + u_{m,4,\alpha}). \end{aligned} \quad (3)$$

Inserting these expansions into $H_l (l=1-4)$, see Table 1) yields the effective Hamiltonian as

$$H_l = \sum_{m,m',\tau,\kappa,\alpha,\beta} J'_{m\tau m'\kappa,\alpha\beta} u_{m,\tau,\alpha} u_{m',\kappa,\beta} \quad (4)$$

where $J'_{m\tau m'\kappa,\alpha\beta}$ – a function of O_γ^W , m , m' , κ , τ , α , and β – characterizes the coupling

between $u_{m,\tau,\alpha}$ and $u_{m',\kappa,\beta}$ dipoles. For instance, the $Hf_x^X Hf_y^Z O_x^{Ay}$ term in H_1 implies the coupling between electric dipoles as $\sum_{m,m'}(u_{m,1,x} + u_{m,2,x} - u_{m,3,x} - u_{m,4,x})(u_{m,1,y} - u_{m,2,y} - u_{m,3,y} + u_{m,4,y})O_x^{Ay}$. By this procedure, we re-formulate each H_l ($l = 1-4$) in terms of electric dipole $u_{m,\tau,\alpha}$, and the corresponding $J'_{m\tau m'\kappa,\alpha\beta}$ interaction can be extracted via

$$J'_{m\tau m'\kappa,\alpha\beta} = \frac{\partial^2 H_l}{\partial u_{m,\tau,\alpha} \partial u_{m',\kappa,\beta}} \quad (5)$$

By definition, the eDMI between $\mathbf{u}_i \equiv (u_{i,x}, u_{i,y}, u_{i,z})$ and $\mathbf{u}_j \equiv (u_{j,x}, u_{j,y}, u_{j,z})$ dipoles is $\mathbf{D}'_{ij} \cdot (\mathbf{u}_i \times \mathbf{u}_j)$ with $\mathbf{D}'_{ij} \equiv (D'_{ij,x}, D'_{ij,y}, D'_{ij,z})$ being the eDMI vector (see Refs.^{37, 39}). Expanding $\mathbf{D}'_{ij} \cdot (\mathbf{u}_i \times \mathbf{u}_j)$ results in $D'_{ij,x}(u_{i,y}u_{j,z} - u_{i,z}u_{j,y}) + D'_{ij,y}(u_{i,z}u_{j,x} - u_{i,x}u_{j,z}) + D'_{ij,z}(u_{i,x}u_{j,y} - u_{i,y}u_{j,x})$. By $i \rightarrow m\tau$ and $j \rightarrow m'\kappa$ replacements, such an expansion together with Equations (2)-(5) yield the evaluation of eDMI strength as

$$A'_{m\tau m'\kappa,\alpha\beta} = \frac{1}{2} (J'_{m\tau m'\kappa,\alpha\beta} - J'_{m\tau m'\kappa,\beta\alpha}) \quad (6)$$

where $D'_{m\tau m'\kappa,x} = A'_{m\tau m'\kappa,yz}$, $D'_{m\tau m'\kappa,y} = A'_{m\tau m'\kappa,zx}$, and $D'_{m\tau m'\kappa,z} = A'_{m\tau m'\kappa,xy}$."

Via this revision, the aforementioned questions are naturally addressed.

- The blue sentences answer the question (A). In particular, the sentence "For instance, the $Hf_x^X Hf_y^Z O_x^{Ay}$ term in H_1 implies the coupling between electric dipoles as $\sum_{m,m'}(u_{m,1,x} + u_{m,2,x} - u_{m,3,x} - u_{m,4,x})(u_{m,1,y} - u_{m,2,y} - u_{m,3,y} + u_{m,4,y})O_x^{Ay}$." demonstrate how to insert the Hf_x^X and Hf_y^Z expansions into the H_1 Hamiltonian. This also allow the readers to visualize the mathematical form of some terms in Equation (4).

- The red sentences answer the question (B). To be specific, we show the links between the eDMI vector $\mathbf{D}'_{ij} \equiv (D'_{ij,x}, D'_{ij,y}, D'_{ij,z})$ and our proposed $A'_{m\tau m'\kappa,\alpha\beta}$ components. We also indicate that $D'_{ij,x}(u_{i,y}u_{j,z} - u_{i,z}u_{j,y}) + D'_{ij,y}(u_{i,z}u_{j,x} - u_{i,x}u_{j,z}) + D'_{ij,z}(u_{i,x}u_{j,y} - u_{i,y}u_{j,x})$ together with Equations (2)-(5) yield Equation (6). This will help readers to understand the derivation of Equation (6).

Thanks to point (2) of the Reviewer #3 and point (1) of the Reviewer #1, we have substantially revised our manuscript, in order to be beneficial to a broader audience. We believe that our revisions will be helpful for readers to understand the logic of this work in an easier way.

(3) “Then, they claim in their abstract (i) We find that various phases of hafnia showcase non-collinear alignment of electric dipoles, which is interpreted by our phenomenological theories. And (ii) This gives evidence that hafnia simultaneously accommodates anti-symmetric and anisotropic symmetric exchange interactions between electric dipoles. But in fact, it is well known that the different phases of Hafnia exhibit various non-collinear arrangements of dipoles (as it is true also for other ferroelectrics: *i.e.*, different phases are associated to different arrangements).”

Authors’ response:

We agree that “different phases are associated to different arrangements”. For example, bulk BaTiO₃ exhibits *P4mm*, *R3m*, *Amm2* polar phases, with the polar displacements being along *z*, *x+y+z* and *x+y* directions, respectively. However, the displacement patterns in these phases are all collinear. In the nano-structured ferroelectrics such as BaTiO₃-SrTiO₃ nanocomposites, the polar displacements may be non-collinear, but such non-collinear dipole patterns are mostly driven by the depolarizing field or domain wall instead of the eDMI. As mentioned in our “Introduction”, materials with non-collinear dipole patterns that are driven by eDMI and/or eASEI are rather rare in nature – identifying such materials being one of our targets.

By symmetry analysis and first-principles calculations, we find that HfO₂ showcases non-collinear dipole patterns that are driven by eDMI. We understand that the non-collinear dipole patterns in hafnia may be well known by Reviewer #3 and few readers. Yet, such a fact is not obvious to the broad audiences (*e.g.*, in the ferroelectric community), to the best of our knowledge. More than that, the origin of the non-collinear dipole patterns in hafnia is elusive as well. Our work sheds light on the role of eDMI in the creation of the non-collinear dipole patterns in hafnia. We therefore believe that our discoveries will be beneficial to researchers working in the area of ferroelectrics.

(4) “Then they claim that it gives evidence it accommodates anti-symmetric and anisotropic symmetry exchange: this is also implicitly known since it is the translation of the first evidence in the language of magnetism.”

Authors’ response:

This is an interesting point. As shown in our “Introduction”, various intriguing magnetic textures (*e.g.*, magnetic vortices, skyrmions and merons) are due to the mDMI and mASEI. In sharp contrast, most of the recently-discovered electric textures (*e.g.*, electric vortices, skyrmions and merons) are either from the depolarizing field or from the domain wall. (Please note that the eDMI-driven non-collinear dipole alignments are rather rare.) Therefore, a direct “translation” from magnetism to electricity seems ambiguous, because our aforementioned alike phenomena (*e.g.*, magnetic skyrmions *versus* electric skyrmions) share different origins.

In our work, we identify the HfO₂ as a representative case which shows non-collinear dipole patterns – driven by the eDMI. Strikingly, our discovery provides an example supporting the “translation” between the magnetic and electric regimes. That is why we think that “our work can deepen the current knowledge of electromagnetism in condensed matter systems such as ferroelectrics, magnets and multiferroics” as mentioned in the “Discussion” section of the Main Text. Furthermore, our work suggests the possibility of realizing intriguing electric textures in hafnia-related materials, being also of potential interest to the ferroelectric community.

In response to this comment, we wrote “Our findings can hopefully deepen the current knowledge of electromagnetism in condensed matter, and imply the possibility of discovering novel states of matter (*e.g.*, electric skyrmions) in hafnia-related materials.” in the abstract of our Main Text and “In other words, HfO₂ and related materials [*e.g.*, (Hf, Zr)O₂ and Y-doped HfO₂] may be ideal candidates to explore novel electric topological textures. Besides, we hope that our work can deepen the current knowledge of electromagnetism in condensed matter systems such as ferroelectrics, magnets and multiferroics.” in the “Discussion” part of the Main Text.

(5) “What might be new is how to connect the language of interatomic interactions into that of magnetism (anti-symmetric and anisotropic symmetry exchange). Here, their message converges to the key role of trilinear couplings but, which by itself is not new again since it has already been mentioned before (which has likely inspired and motivated their work!).”

Authors’ response:

This comment motivates us to explore more deeply the eDMI and eASEI in hafnia, taking advantage of the interatomic interactions. This yields some conclusions beyond those (obtained by analyzing the trilinear couplings) in our previous version of manuscript. To be specific, we start from the H_1 , H_2 , H_3 and H_4 that capture the non-collinear alignments of dipoles in hafnia, and consider a generalized Hamiltonian $H = \alpha H_1 + \beta H_2 + \gamma H_3 + \delta H_4$.

(i) Working with the Hamiltonian H , we extract the eDMI and eASEI in hafnia contributed by various O_Y^W distortions. We summarize the results in Table 2 (Main Text) and the Supplementary Note 4 (Supplementary Information). **The detailed analysis indicates that the eDMI drives the non-collinear dipole alignments in hafnia while the eASEI is not relevant to such alignments.**

(ii) We compute the interatomic force constants (between two Hf ions) in hafnia without structural distortion or with a O_x^{Ay} distortion. (Please note that the interatomic force constants are not only driven by structural distortions but also contributed by the long-range/short-range dipolar interactions.) The results are shown in Table 3 of the Main Text. We find that the long-range and short-range dipolar interactions also contribute to the eASEI, even in the absence of O_Y^W distortion.

In this response to this comment, we wrote “**The non-null $A'_{12,xy} = -A'_{12,yx} = \alpha_2 O_x^{Ay}$ (respectively, $A'_{14,xy} = -A'_{14,yx} = \alpha_1 O_x^{Ay}$) components of eDMI imply the non-collinear alignments of electric dipoles – within the xy plane – centered on Hf₁ and Hf₂ (respectively, Hf₁ and Hf₄) sites, coinciding with Figs. 3b and 3e. The detailed analysis confirms that the eDMI drives the NCDP in the $P2_1/c$, $Pmn2_1$, $Pca2_1$ and $Pbca$ phases of HfO₂ (see Supplementary Note 4 of the SI)” in page 5 of the Main Text, and “**Our detailed analysis, as shown in Supplementary Note 4 the SI, implies that the eASEI being hosted by $H = \alpha H_1 + \beta H_2 + \gamma H_3 + \delta H_4$ is not relevant to****

the NCDP in the $P2_1/c$, $Pmn2_1$, $Pca2_1$ and $Pbca$ phases of HfO_2 .” in page 6 of the Main Text. We also added a section entitled “The long-range and short-range interactions” in page 6 of the Main Text and a “Supplementary Note 4” in the Supplementary Information.

(6) “In the present work, it is indeed more interestingly reformulated, in a more practical way, but in itself, this result is rather incremental and this reformulation is finally not fully exploited (i.e. they do not report anything quantitative). What would be a more significant step forward is to use this reformulation to compute coefficients and exemplify quantitatively the role of the different terms. In fact I am questioning why, in spite of what is reported in Fig. 2, they did not extract and report quantitative parameter !? Are all terms contributing evenly or is there one dominating?”

Authors’ response:

This comment is excellent! Following this point, we now do a quantitative analysis regarding Fig. 2 and find that different terms may contribute unequally. Regarding this, we wrote “For instance, the fittings in Fig. 2b – with the R^2 (i.e., coefficient of determination) exceeding 0.999 – indicate the linear dependence of Hf_y^Y and Hf_x^Z on Hf_z^F . The fitting slopes of $14.58 \text{ eV \AA}^{-2}$ for $Hf_z^F Hf_y^Y O_z^{Ay}$ and $11.04 \text{ eV \AA}^{-2}$ for $Hf_z^F Hf_x^Z O_x^{Az}$ show that the $Hf_z^F Hf_y^Y O_z^{Ay}$ and $Hf_z^F Hf_x^Z O_x^{Az}$ terms contribute unequally in HfO_2 .” in page 3 of the Main Text.

Furthermore, the exchange interactions between electric dipoles (including the eDMI and eASEI) can be quantitatively characterized by the interatomic force constants [see the “Authors’ response” to the point (2) of the Reviewer #2]. In “The O_γ^W -contributed anti-symmetric exchange interactions” section, we quantitatively assess the eDMI in HfO_2 (see Fig. 4 and the corresponding discussion in page 5 of the Main Text). We find that the eDMI are solely contributed by the O_γ^W -type distortion and show almost-perfect linear relationship with O_γ^W . The eDMI between different pairs of Hf ions may depend on O_γ^W unequally. In this regard, we wrote “The fittings also yield that $\alpha_1 = -0.99 \text{ eV \AA}^{-3}$ and $\alpha_2 = 3.58 \text{ eV \AA}^{-3}$, implying the unequal dependences of $A'_{12,xy}$ and $A'_{14,xy}$ on O_x^{Ay} ” in page 5 of the Main Text.

We also provide other quantitative discussion regarding the exchange interactions in hafnia [for details, please see the “Authors’ response” to the point (7) of the Reviewer #3].

(7) “Also how does those trilinear terms compare to other terms in the energy? It should be noticed that other kinds of terms (than trilinear) could contribute to J and then A and S and there is no guarantee that focusing on the trilinear ones properly capture the dominant part. This has to be done applying a quantitative model up to the end ...”

Authors’ response:

This is another excellent comment. The exchange interactions between electric dipoles can be characterized by the interatomic force constants, if we use the displacements of the ferroelectrically-active ions to represent the electric dipoles. In such sense, the exchange interactions between electric dipoles can be formulated as

$$H = \sum_{i \neq j} J_{ij}^{\text{iso}} \mathbf{u}_i \cdot \mathbf{u}_j + \sum_{i \neq j, \alpha \neq \beta} A'_{ij, \alpha\beta} (u_{i, \alpha} u_{j, \beta} - u_{i, \beta} u_{j, \alpha}) + \sum_{i \neq j} S'_{ij, \alpha\beta} u_{i, \alpha} u_{j, \beta}$$

[Equation (9) in the Main Text]. Here, the exchange interactions between electric dipoles include the Heisenberg-like, the anti-symmetric, and the anisotropic symmetric terms, similar to the cases in magnetism [see Equation (8) in the Main Text]. Definitely, other kinds of contributions (*e.g.*, the long-range and short-range dipolar interactions) coexist with those arising from the trilinear couplings. To clarify this point, we wrote “In this formula, the $A'_{ij, \alpha\beta}$ and $S'_{ij, \alpha\beta}$ are the eDMI and eASEI parameters. This time, the $A'_{ij, \alpha\beta}$ and $S'_{ij, \alpha\beta}$ parameters may be contributed by various factors such as structural distortions but also long-range and short-range dipolar interactions (*i.e.*, not solely by structural distortions). This is readily clarified by comparing \tilde{J}'_{13} with \tilde{J}''_{13} (see Table 3).” in page 7 of the Main Text. In this regard, we also added a section entitled “Exchange interactions: magnetic versus electric” in page 7 of the Main Text.

We move on to explore the contribution of these trilinear couplings in the exchange interactions. To this end, we compute the exchange interactions between electric dipoles for (i) the high-symmetric $Fm\bar{3}m$ phase with null structural distortion, and (ii) the $Fm\bar{3}m$ phase superimposed by

a O_x^{Ay} distortion. The results are summarized in Table 3 of the Main Text.

For case (i), the exchange interactions between Hf₁ and Hf₂ ions are given by

$$\tilde{J}'_{12} = \begin{pmatrix} 3.30 & 0.00 & 0.00 \\ 0.00 & -2.24 & -2.67 \\ 0.00 & -2.67 & -2.24 \end{pmatrix}.$$

Here, diagonal components $\tilde{J}'_{12,xx}$, $\tilde{J}'_{12,yy}$ and $\tilde{J}'_{12,zz}$ are 3.30, -2.24 and -2.24 eV Å⁻², respectively. Furthermore, there are also two off-diagonal components, namely, $\tilde{J}'_{12,yz} = \tilde{J}'_{12,zy} = -2.67$ eV Å⁻². These interactions are not relevant to the non-collinear alignments of electric dipoles. (Please note that these interactions are not related to the O_y^W -type distortions.) Meanwhile, the exchange interactions between Hf₁ and Hf₃ ions are characterized by \tilde{J}'_{13} , shown below. In $Fm\bar{3}m$ phase, the \tilde{J}'_{13} is linked with \tilde{J}'_{12} by symmetry (e.g., $\tilde{J}'_{13,yy} = \tilde{J}'_{12,xx} = 3.30$ eV Å⁻²).

$$\tilde{J}'_{13} = \begin{pmatrix} -2.24 & 0.00 & -2.67 \\ 0.00 & 3.30 & 0.00 \\ -2.67 & 0.00 & -2.24 \end{pmatrix}$$

For case (ii), the O_x^{Ay} distortion modifies \tilde{J}'_{12} to \tilde{J}''_{12} , where \tilde{J}''_{12} is given by

$$\tilde{J}''_{12} = \begin{pmatrix} 3.07 & 0.88 & 0.73 \\ -0.88 & -2.48 & -2.92 \\ -0.73 & -2.92 & -2.45 \end{pmatrix}.$$

In \tilde{J}''_{12} , the eDMI are arising from the O_x^{Ay} distortion and characterized by $\tilde{J}''_{12,xy} = -\tilde{J}''_{12,yx} = 0.88$ eV Å⁻² and $\tilde{J}''_{12,xz} = -\tilde{J}''_{12,zx} = 0.73$ eV Å⁻². The $\tilde{J}''_{12,xy}$ and $\tilde{J}''_{12,xz}$ interactions, being responsible for the non-collinear dipole patterns in hafnia, are about 24% of the $\tilde{J}'_{12,xx}$ interaction.

Besides, the O_x^{Ay} distortion modifies \tilde{J}'_{13} to \tilde{J}''_{13} with \tilde{J}''_{13} given by

$$\tilde{J}''_{13} = \begin{pmatrix} -2.12 & 0.25 & -2.67 \\ 0.25 & 3.24 & 0.06 \\ -2.67 & 0.06 & -2.26 \end{pmatrix}.$$

By comparing \tilde{J}''_{13} with \tilde{J}'_{13} , we find that the O_x^{Ay} distortion induces anisotropic symmetric components $\tilde{J}''_{13,xy} = \tilde{J}''_{13,yx} = 0.25$ eV Å⁻² and $\tilde{J}''_{13,yz} = \tilde{J}''_{13,zy} = 0.06$ eV Å⁻². The $\tilde{J}''_{13,xy}$ interaction is about 8% of the $\tilde{J}'_{13,yy}$ interaction.

The detailed discussion is available in the section “The long-range and short-range interactions” of the Main Text.

(8) “In conclusion, I consider it is a potentially very interesting paper but, reaching its end, I am a bit disappointed. I have the feeling that they stopped half-way. They propose an interesting approach; now they should apply it to quantify things better if they really want to make a step forward. For these reasons, I am not recommending this manuscript at this stage.”

Authors' response:

We thank Review #3 for raising so many constructive comments. As shown above, we have made enormous efforts to address these comments. We hope that our revised manuscript can now be recommended for publication in *Nature Communications*.

REVIEWERS' COMMENTS

Reviewer #1 (Remarks to the Author):

The authors have provided a substantial effort for improving the presentation and readability of their study. As I mentioned at the first round, this study opens a new array for solid state physics with the identification anti-symmetric and anisotropic symmetric exchange interactions between electric dipole (that are analogous to interactions between magnetic dipoles). This could yield several applications in electronics such as electrical skyrmions.

I recommend publication of the manuscript in Nature Communications.

Reviewer #2 (Remarks to the Author):

In this revised version, the authors have well addressed my previous concerns and done proper revisions. I'm satisfied by this revised manuscript and glad to recommend the acceptance.

Reviewer #3 (Remarks to the Author):

In their reply, the authors have convincingly addressed all my original criticisms. They have implemented appropriate changes in the revised manuscript. At this stage, I consider the paper as much more robust and convincing and I am glad to recommend it for publication.

Response Letter

We thank Reviewers #1, #2 and #3 for recommending our manuscript for publication. Our point-by-point responses to the reviewers' comments are given below.

Reviewer #1 has the following comments:

The authors have provided a substantial effort for improving the presentation and readability of their study. As I mentioned at the first round, this study opens a new array for solid state physics with the identification anti-symmetric and anisotropic symmetric exchange interactions between electric dipole (that are analogous to interactions between magnetic dipoles). This could yield several applications in electronics such as electrical skyrmions.

I recommend publication of the manuscript in Nature Communications.

Authors' response:

We appreciate it that Reviewer #1 evaluates our work as “this study opens a new array for solid state physics” and “this could yield several applications in electronics such as electrical skyrmions”. We thank Reviewer #1 for recommending our work for publication.

Reviewer #2 has the following comments:

In this revised version, the authors have well addressed my previous concerns and done proper revisions. I'm satisfied by this revised manuscript and glad to recommend the acceptance.

Authors' response:

We thank Reviewer #2 for recommending our work for publication.

Reviewer #3 has the following comments:

In their reply, the authors have convincingly addressed all my original criticisms. They have implemented appropriate changes in the revised manuscript. At this stage, I consider the paper as much more robust and convincing and I am glad to recommend it for publication.

Authors' response:

We thank Reviewer #3 for recommending our work for publication.